# Selective Deferred Routing: Enabling Cost-Efficient Collaboration Between Local SLMs and Remote LLMs

**Qijun Miao** [1] [2]  **Zhixuan Fang** [1] [3]

## Abstract

The rapid advancement of large language models (LLMs) has led to remarkable performance across diverse domains, making them indispensable assistants in daily life and work. Currently, LLM services are primarily accessed in two ways: (i) paid access to cloud-hosted LLMs, which are powerful but introduce nontrivial cost; and (ii) deployment of small language models (SLMs) on personal devices or small clusters, which, while less powerful, are sufficient for handling relatively simple tasks. To achieve a balanced trade-off between monetary cost and task performance, we propose Selective Deferred Routing, a paradigm that enables cost-efficient collaboration between local SLMs and remote LLMs. In this framework, a user request is first processed by the local SLM, which not only generates a preliminary response but also provides rich semantic representations of the request. A lightweight decision module then leverages this information to either adopt the initial response or route the request to the most suitable remote LLM for a higher-quality response. Extensive experiments across diverse model architectures and families, including both SLMs and LLMs, as well as datasets spanning multiple task scenarios, demonstrate that our approach consistently outperforms existing multi-LLM collaboration methods under a wide range of cost–performance trade-offs.

## 1. Introduction

Recent advancements in pre-trained language models based on the Transformer architecture (Vaswani et al., 2017) have demonstrated remarkable capabilities across a wide range of tasks, including question answering, creative writing, programming, and even complex reasoning (Brown et al., 2020). Consequently, Large Language Models (LLMs) are increasingly used as indispensable assistants in both everyday and professional settings.

Nowadays, most end-users obtain LLM services via two approaches. The first is paid access to cloud-hosted LLMs: very large models running on massive remote compute infrastructure that offer the strongest capabilities but incur nontrivial monetary cost. The second is local deployment: running relatively smaller LLMs on personal devices or on small on-premises clusters, enabling essentially cost-free access to inference at the point of use. Although smaller models are typically less capable than large cloud models, they are often sufficient for routine tasks that do not require extensive reasoning or access to very large knowledge bases (Chen & Varoquaux, 2024; Subramanian et al., 2025).

On the other hand, user demands for LLM services are highly diverse and span different levels of complexity, with a substantial portion concerning everyday advice and assistance (Chatterji et al., 2025). Consequently, *dynamically choosing or combining the above two LLMs using approaches according to the task characteristics offers a promising way to trade off monetary cost and task performance*. For clarity and brevity, in the remainder of this paper, we refer to paid, cloud-hosted LLMs as remote LLMs, and to smaller LLMs hosted on personal devices or small on-premises clusters as local SLMs. Moreover, considering the computational resource limitations of individuals and small institutions, we assume that there is always one local SLM in a collaboration system.

Focusing on cost-efficient multi-LLM collaboration with respect to task characteristics, existing studies have primarily explored two paradigms. The first is *Immediate Routing* (Ong et al.; Ding et al., 2024; Hu et al., 2024; Ding et al., 2025; Song et al., 2025). As shown in Figure 1 (a), a trained performance prediction model takes the user's prompt as input to forecast the performance of each candidate LLM. Then, a term measuring the estimated LLM inference cost will be subtracted from the predicted performance. The model with the highest resulting score is selected to handle

[1]Institute for Interdisciplinary Information Sciences, Tsinghua University, Beijing, China [2]Xiongan AI Institute, Xiong'an, China [3]Shanghai Qi Zhi Institute, Shanghai, China. Correspondence to: Zhixuan Fang <zfang@mail.tsinghua.edu.cn>.

*Proceedings of the 43rd International Conference on Machine Learning*, Seoul, South Korea. PMLR 306, 2026. Copyright 2026 by the author(s).

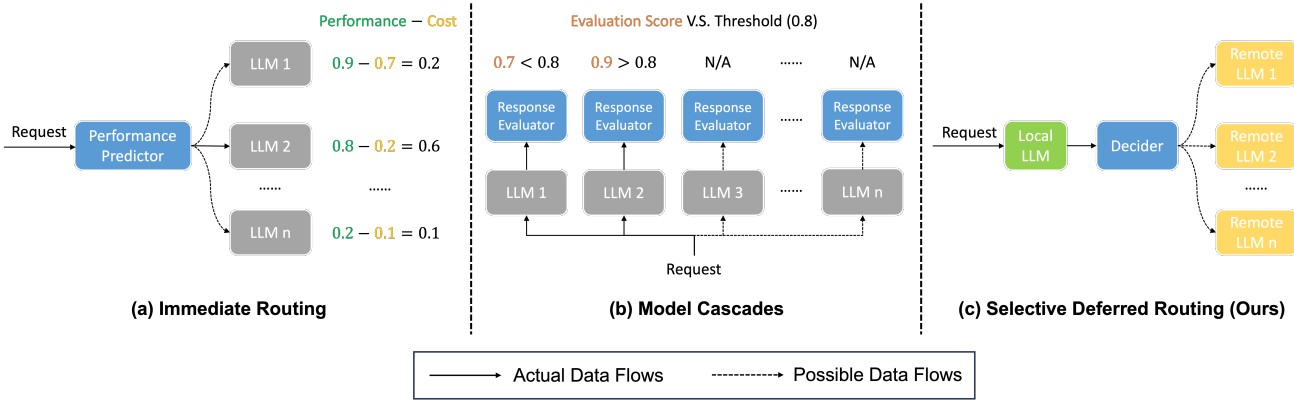

*Figure 1.* The workflow overview of two existing multi-LLM collaboration paradigms (Immediate Routing and Model Cascades) and the proposed Selective Deferred Routing.

the user request. The workflow is straightforward and intuitive; however, because it relies solely on the user prompt and lacks additional decision-making cues, its performance in practice is often limited.

The second paradigm is *Model Cascades* (Chen et al., 2024; Aggarwal et al., 2024; Yue et al., 2024; Nie et al., 2024; Zhang et al., 2024b; Dekoninck et al., 2025). As shown in Figure 1 (b), it organizes all available LLMs into a sequential chain, with the assumption that LLMs positioned later in the chain possess greater capabilities but also incur higher costs. The system invokes models along this cascade to handle the user request, then evaluates their response using a trained model. If any evaluation result exceeds a predefined threshold, the response will be returned early to the user without invoking the remaining LLMs. This paradigm stems from the speculative notion that low-cost models may suffice to handle a given task. However, its linear structure can lead to the invocation of all available LLMs in the worst case, resulting in substantial costs and increased response latency. Moreover, the assumption of a strict price–performance ordering among LLMs does not hold in today's heterogeneous model market.

It is also noteworthy that both of the paradigms discussed above treat all available LLMs as black boxes. In the practical scenarios considered in this paper, however, local SLMs can provide information beyond textual responses (e.g., hidden states), which can be leveraged to inform later decisions. Another line of research has focused on the characteristics of local SLMs and remote LLMs and their collaboration (He et al., 2024; Zhang et al., 2024a; Hao et al., 2024; Liu et al., 2025; Xu et al., 2024), but primarily emphasizes system-level metrics such as latency, throughput, and load balancing, rather than the trade-off between monetary cost and task performance.

Based on the above observations and analysis, we propose

Selective Deferred Routing (SDR), a paradigm that enables cost-efficient collaboration between local SLMs and remote LLMs. As shown in Figure 1 (c), it initially delegates user requests to the local SLM, which not only produces a textual response but also provides rich feature representations of the task. A lightweight decision module then utilizes the enriched information to make a selective routing decision, in which those already satisfactory responses from the local SLM will be directly returned, while the others will be routed to the most suitable remote LLM for a higher-quality response.

To obtain a sufficiently powerful decision module, we begin with a simplified yet fundamental formulation, the Local-Remote Collaboration problem, that involves only one remote LLM. In this setting, the selective routing problem can be essentially formulated as an AUC (Area Under Curve) optimization problem, which yields a more effective training objective than naive quality prediction. Then, we design the decision module by reusing the architecture of a single transformer block from the local LLM. This allows the decision module to maximally leverage the token-wise hidden states produced by the SLM while remaining lightweight. Finally, this design can be efficiently extended to scenarios with multiple remote LLMs.

The main contributions of this work are summarized below:

- We propose Selective Deferred Routing (SDR), a novel collaboration paradigm between local SLMs and remote LLMs, which can achieve a more effective balance between cost and performance.

- We build a strong decision module by analyzing the Local-Remote Collaboration problem, which leads to a more effective training objective. Our decision module design is lightweight yet effective, and can be further extended to scenarios with multiple remote LLMs.

- Extensive experiments across diverse model architectures and families, as well as datasets spanning multiple task scenarios, demonstrate that our approach consistently outperforms existing multi-LLM collaboration methods under a wide range of cost–performance trade-offs. Our code is available at `https://github.com/miaoqijun/SDR`.

## 2. Related Work

### 2.1. Cost-Efficient Multi-LLM Collaboration

A substantial body of recent work has demonstrated interest in collaborative systems involving multiple LLMs with varying costs and performance characteristics. As shown in Figure 1, *Immediate Routing* relies solely on the user prompt to predict the performance of all available LLMs. Hybrid LLM (Ding et al., 2024) fine-tunes a BERT-based model to predict the performance of two available LLMs for a given user request. RouteLLM (Ong et al.) also focuses on scenarios with two available LLMs and leverages human preference labels between them from Chatbot Arena (Chiang et al., 2024). It evaluates three training strategies, including similarity-weighted ranking, matrix factorization, and a BERT classifier, to learn effective routing between the larger and smaller LLMs. RouterBench (Hu et al., 2024) employs Sentence Transformers (Reimers & Gurevych, 2019) to obtain embeddings of the user prompt, followed by the application of KNN and MLP models to map these embeddings to all available LLMs' performance.

Another collaboration paradigm, *Model Cascades*, organizes all available LLMs into a predefined ordered cascade, then progressively invokes models until a sufficiently satisfactory response is obtained. FrugalGPT (Chen et al., 2024) fine-tunes a BERT-based model to evaluate the quality of LLM responses. Mixture of Thought (Yue et al., 2024) leverages multiple sampling: for each user request, multiple responses are sampled from a single LLM, and the consistency among these responses is used to estimate the model's confidence, which in turn acts as a proxy for response quality. AutoMix (Aggarwal et al., 2024) attempts to have LLMs provide a confidence score alongside their answers, which serves as a proxy for response quality, while also allowing the system to adaptively skip intermediate models in the cascade. Dekoninck et al. (2025) further generalizes cascade routing by dynamically selecting candidate models within a predefined ordered model pool instead of strictly traversing the cascade sequentially. In addition, some work focuses on more specialized scenarios, such as online learning on streaming data (Nie et al., 2024), reinforcement learning under budget-constrained settings (Zhang et al., 2024b), etc.

As noted in Section 1, these methods treat all available LLMs as black boxes, considering only their textual inputs and outputs. In practical scenarios, however, local SLMs can provide richer information that can be leveraged for decision-making. Moreover, the workflow and organizational structure of these methods do not align well with the characteristics of today's heterogeneous model market or with user requirements.

### 2.2. System Design between Local SLMs and Remote LLMs

Another line of research focuses on leveraging the distinct characteristics of local SLMs and global LLMs for system-level design. Computational offloading (He et al., 2024; Zhang et al., 2024a; Hao et al., 2024) aims to schedule tasks across devices based on their computational resources, workloads, and other characteristics, in order to optimize system-level metrics such as load balancing and user-side latency. Edge–cloud collaborative speculative decoding (Liu et al., 2025; Xu et al., 2024) treats local SLM as draft models while performing token validation on the remote LLM, thereby increasing system throughput without compromising performance. In general, although these studies also recognize the unique value of local SLMs in collaborative systems, they primarily focus on optimizing service-level metrics such as latency, throughput, and load balancing, rather than balancing task performance and monetary cost, making them a parallel line of research to our work.

## 3. Selective Deferred Routing

In this section, we present the design and implementation details of Selective Deferred Routing. In Section 3.1, we begin with a simplified yet fundamental setting, the Local-Remote Collaboration problem, which involves the local SLM and one remote LLM. It allows for a more principled analysis under a well-structured formulation in Section 3.2. In Section 3.3, we provide the design details of our decision module. In Section 3.4, we extend the design to the scenario with multiple remote LLMs, providing a practical solution that better aligns with real-world user demands.

### 3.1. Local-Remote Collaboration Problem

We begin with a simplified yet fundamental setting. Consider one local SLM, denoted as $M_S$, and one remote LLM, denoted as $M_L$. Each model $M \in \{M_S, M_L\}$ takes a user query $q_i \in \mathcal{Q}$ as input and generate an answer $a_i \in \mathcal{A}$. We also have a dataset $\mathcal{D} = (\mathcal{Q}, p)$, consisting of a query set $\mathcal{Q} = \{q_i\}_{i=1}^N$ of size $N$ and a performance evaluator $p : (\mathcal{Q}, \mathcal{A}) \to [0, 1]$ measuring the quality of an answer to the given query. The implementation of $p$ depends on the dataset. For datasets with binary labeling, we set $p = 1$ for correct answers and $p = 0$ otherwise. For datasets with graded evaluation metrics, we can normalize the resulting metrics to the range $[0, 1]$ to ensure consistency across

datasets.

Our routing problem in this simplified setting reduces to a binary choice: either accept the local SLM's answer directly or forward the query to the remote LLM. We define the input information required for routing decisions as $\mathcal{X} = \{x_i\}_{i=1}^N$. As discussed in Section 1, the specific content of this information depends on the design of the system workflow: it may consist solely of the user input $q_i$, include the output answer $a_i$ generated by the local SLM, or further incorporate intermediate representations produced during the SLM's inference process. The result can be obtained following the binary classification paradigm, in which we first define a learnable scoring model, parameterized by $\theta$, as a function $f_\theta : \mathcal{X} \to \mathbb{R}$. For a given input $x_i$, it predicts a continuous scalar score $s_i = f_\theta(x_i)$, which quantifies the preference for accepting the local SLM's answer. The final decision is determined by comparing the score $s$ against a decision threshold $\alpha \in \mathbb{R}$. The *binary selective routing* (BSR) result $\mathrm{BSR}_i(\alpha) \in \{M_S, M_L\}$ is defined as:

$$\mathrm{BSR}_i(\alpha) = \begin{cases} M_S, & \text{if } s_i \geq \alpha \\ M_L, & \text{otherwise.} \end{cases} \tag{1}$$

For a fixed threshold $\alpha$, we can derive the overall cost and task performance of the system over the dataset $\mathcal{D}$. For clarity and practical relevance, we measure the overall user's cost as the proportion of queries that are routed to the remote LLM:

$$\mathrm{Cost}(\alpha) = \frac{1}{|N|} \sum_{i=1}^N \mathbf{1}\{\mathrm{BSR}_i(\alpha) = M_L\}. \tag{2}$$

Then we define the overall task performance as the average output quality of the LLMs that the queries are finally routed to:

$$\mathrm{Perf}(\alpha) = \frac{1}{|N|} \sum_{i=1}^N p\left(q_i, \mathrm{BSR}_i(\alpha)(q_i)\right). \tag{3}$$

To isolate the impact of the router, we can capture the relative improvement by rewriting $\mathrm{Perf}(\alpha)$ in terms of the *performance gain* (PG):

$$\mathrm{PG}(\alpha) = \frac{\mathrm{Perf}(\alpha) - \lim_{\alpha_0 \to -\infty} \mathrm{Perf}(\alpha_0)}{\lim_{\alpha_1 \to +\infty} \mathrm{Perf}(\alpha_1) - \lim_{\alpha_0 \to -\infty} \mathrm{Perf}(\alpha_0)}, \tag{4}$$

in which $\lim_{\alpha_0 \to -\infty} \mathrm{Perf}(\alpha_0)$ and $\lim_{\alpha_1 \to +\infty} \mathrm{Perf}(\alpha_1)$ represent the overall performance of the local SLM and that of the remote LLM, respectively.

### 3.2. AUC Optimization: Metric and Method

As the threshold $\alpha$ sweeps from $+\infty$ to $-\infty$, the system traverses a curve in the two-dimensional Cost-Performance plane. To evaluate the BSR module while applying different

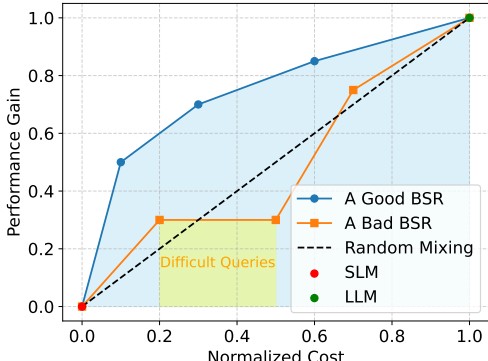

*Figure 2.* An example Cost-Perf Curve obtained from two possible BSR modules with several sampled $\alpha$.

$\alpha$, we adopt the Area Under the Cost-Performance Curve (AUC) as the optimization objective. By treating the average performance $P(\alpha)$ and cost $C(\alpha)$ as a parametric curve indexed by the threshold $\alpha$, the AUC is formulated as:

$$\mathrm{AUC}(\theta) = \int_0^1 \mathrm{PG}\left(\alpha\right) \mathrm{d}\left(\mathrm{Cost}(\alpha)\right). \tag{5}$$

This metric serves as the primary evaluation criterion for routing methods, as it captures the overall expected utility across users with different budget constraints. In addition, we provide a simple and practical procedure for selecting an appropriate threshold given a user-specified budget in Appendix B.

Figure 2 provides an example curve for better understanding. Intuitively, a BSR module achieving a large AUC (the blue curve) directly indicates that users can obtain performance close to that of the remote LLM even under a limited budget.

Next, we aim to train the parameterized scoring model to optimize the cost-performance AUC. A straightforward proxy objective is to predict the quality of the local SLM's answer, which is widely adopted in model cascades (Chen et al., 2024; Aggarwal et al., 2024; Yue et al., 2024). Answers with sufficiently high predicted quality (above the threshold) are returned directly, while queries with low-quality answers are routed to the remote LLM in the hope of obtaining a higher-quality response.

While this approach is intuitive, it does not effectively optimize the AUC metric in practice. For instance, consider a particularly difficult query for which neither the local SLM nor the remote LLM can generate a high-quality answer. In such cases, the optimal choice is to return the local answer directly to save cost. However, under the aforementioned training paradigm, as shaded in the yellow area of Figure 2, the decision module could still be biased toward routing the query to the remote LLM with a relatively low $\alpha$, leading to

unnecessary expense without improving performance.

From the above example, we draw a simple but important insight: in binary selective routing, the decisive factor is not the absolute performance of either the SLM or the LLM, but rather their *performance gap*. We define this gap for a given query $q_i$ as:

$$\text{Gap}_i = p(q_i, M_L(q_i)) - p(q_i, M_S(q_i)), \forall i \in N. \quad (6)$$

We then state the following result:

**Theorem 3.1.** *The routing decisions produced by the BSR module yield the optimal cost–performance AUC if and only if $\forall i, j \in N$,*

$$Gap_i \geq Gap_j \implies s_i \leq s_j.$$

We defer the detailed proof to Appendix A and instead provide an intuitive explanation here. As the threshold $\alpha$ increases, queries are progressively routed to the remote LLM in ascending order of their scores. Queries with larger performance gaps contribute steeper slopes on the cost–performance curve. Consequently, routing such high-gap queries to the remote LLM earlier, i.e., toward the left side of Figure 2, yields a more convex curve, thereby resulting in a higher AUC.

To satisfy the conditions required by Theorem 3.1, we train the scoring model in a supervised manner to act as a linear regressor that predicts the negative of the per-query performance gap, which leads to the following training objective:

$$\mathcal{L} = \frac{1}{B} \sum_{i=1}^{B} \left( -\text{Gap}_i - s_i \right)^2, \quad (7)$$

where $B$ denotes the batch size. To ensure generality and simplicity, we adopt the standard mean squared error (MSE) loss for training. In addition, we experiment with several alternative training objectives that incorporate ranking information, which better reflect the theoretical conditions imposed by Theorem 3.1. The corresponding results and analyses are reported in Appendix F.1.

### 3.3. Scoring Model Design

We now describe the details of the scoring model, which is designed to fully utilize the token-wise information produced by the local SLM while remaining lightweight and practical for on-device deployment.

As illustrated in Figure 3, the scoring model consists of a single transformer layer, which adopts the same architecture as that of the local SLM, followed by an MLP. By leveraging per-token hidden state outputs, the transformer layer can effectively capture the contextual information of the user query and the local SLM's response. Then, the output of the

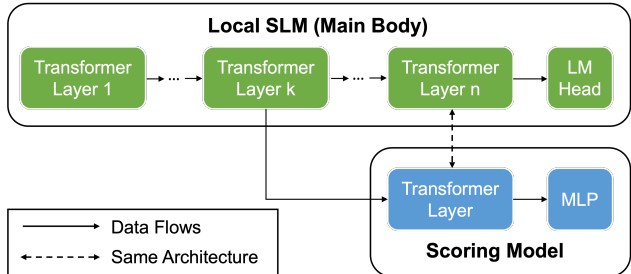

*Figure 3.* The architecture of the scoring model.

last token, which inherently aggregates context from all preceding positions, is extracted to be fed into the subsequent MLP.

We use the hidden states from an intermediate transformer layer of the SLM as input to our model. The choice of the layer depth $k$ is guided by the following considerations. First, a well-established finding in prior work is that the hidden states produced by the final transformer layer of a pretrained LLM tend to be strongly aligned with the pre-training objective, i.e., language modeling, and therefore exhibit limited generalization to other downstream tasks (Van Aken et al., 2019; Skean et al., 2024; 2025). Second, from a practical deployment perspective, with a $k$ smaller than the total number of layers, a higher degree of hardware parallelism is encouraged, which has the potential to reduce the additional computational overhead introduced by the decision module. Finally, we set $k$ to the penultimate layer in Section 4. We also explore alternative design choices, including using different $k$ and employing different scoring model architectures; the corresponding results are reported in Appendix F.2.

Importantly, the scoring model is lightweight and practical for deployment on consumer-grade devices. For instance, when using Qwen-2.5-7B (Qwen Team, 2024) as the local SLM, which comprises 28 transformer layers with approximately 0.233B parameters per layer, the corresponding scoring model's parameter count is orders of magnitude smaller than that of the full local SLM, leading to a minimal memory footprint. More detailed measurements of the additional computational and memory overhead introduced by the decision module are reported in Appendix D.

### 3.4. Towards Collaboration with Multiple Remote LLMs

To further match the real user demands, we extend the aforementioned design from the simplified setting to the multiple remote LLMs setting. Suppose we now have $|M_L|$ remote LLMs, $\{M_L^{(1)}, M_L^{(2)}, \cdots, M_L^{(|M_L|)}\}$. We first need the scoring model to output $|M_L|$ scores corresponding to

each remote LLM by simply modifying the model architecture to fit the output dimensionality $f_\theta : \mathcal{X} \to [0,1]^{|M_L|}$, then train it with the same MSE objective. In the experiments reported in Section 4, we present results obtained by simply varying the dimensionality of the final linear layer of the decision module. In Appendix F.3, we further explore more variants with different degrees of parameter sharing.

To incorporate the pricing of different remote LLMs into the decision making, we estimate the per-request cost for each LLM. Specifically, for the $j$-th remote LLM, the cost of serving a query $q_i$ is defined as:

$$C_i^{(j)} = n_{\text{in}}(q_i) \cdot p_{\text{in}}(M_L^{(j)}) + \hat{n}_{\text{out}}^{(j)}(q_i) \cdot p_{\text{out}}(M_L^{(j)}), \quad (8)$$

where $n_{\text{in}}(q_i)$ denotes the number of input tokens of query $q_i$, $\hat{n}_{\text{out}}^{(j)}(q_i)$ is the predicted number of output tokens for the $j$-th remote LLM[1], and $p_{\text{in}}(M_L^{(j)})$ and $p_{\text{out}}(M_L^{(j)})$ represent the per-token prices for input and output tokens of the $j$-th LLM, respectively.

Next, since the scoring model is trained to align with the negative performance gap between the local SLM and each remote LLM, we formalize the utility of routing a query $q_i$ to the $j$-th remote LLM as:

$$U_i^{(j)}(k) = -s_i^{(j)} - k \cdot C_i^{(j)}, \quad (9)$$

where $s_i^{(j)} = f_\theta(x_i)[j]$ corresponds to the negative performance gap of the $j$-th remote LLM over the local SLM, and $C_i^{(j)}$ represents the estimated cost of invoking the $j$-th remote LLM. The parameter $k$ controls the trade-off between cost efficiency and performance, reflecting the user's preference for lower monetary cost versus higher response quality. By the definition above, the utility of keeping the local SLM response is zero. Finally, we can make the *selective routing* (SR) decision as:

$$\text{SR}_i(k) = \begin{cases} M_S, & \text{if } U_i^{(j)}(k) \leq 0 \;\; \forall j, \\ \underset{M_L^{(j)}}{\arg\max} \; U_i^{(j)}(k), & \text{otherwise.} \end{cases} \quad (10)$$

## 4. Evaluation

In this section, we conduct a comprehensive evaluation of the proposed method in terms of both algorithmic performance and system efficiency. Section 4.1 details the experimental settings, including the LLMs used, the datasets, and the baseline methods. Sections 4.2 and 4.3 report cost-efficiency comparisons between SDR and the baselines under the single-remote and multi-remote settings, respectively.

Section 4.4 conducts controlled ablations to isolate the contribution of each design component. Section 4.5 further examines the practical performance of the local–remote collaborative system and provides a brief analysis of the end-to-end latency. We also provide additional generalization analysis in Appendix F.4 to further assess the practical applicability of the proposed method in realistic deployment scenarios.

### 4.1. Settings

**LLMs.** We adopt Llama-3.1-8B (Meta AI, 2024) and Qwen-2.5-7B (Qwen Team, 2024) as local SLMs, which are widely used in real-world deployments. For remote LLMs, we select Llama-4-Maverick (402B) (Meta AI, 2025), and DeepSeek-V3 (685B) (DeepSeek-AI, 2024) from the family of large-scale open-source models. For proprietary models, we also select GPT-4o (OpenAI, 2024) and o4-mini (OpenAI, 2025) as representatives of general-purpose and reasoning-specialized models, respectively. We leave further details on LLM usage to Appendix C.

**Datasets.** We experiment with a diverse set of datasets covering different aspects of model capabilities: (i) MMLU (Hendrycks et al., 2021): a multiple-choice dataset designed to assess models' broad knowledge and reasoning ability across various domains; (ii) SQuAD (Rajpurkar et al., 2016): a free-form question answering dataset targeting reading comprehension over passages; (iii) GSM8K (Cobbe et al., 2021): a grade-school-level math problem dataset evaluating models' numerical reasoning and problem-solving skills. Given the varying sizes of these datasets, we adopt different splits for the experiments while ensuring that the same train–test partitions are used across both our proposed method and all baseline methods. More detailed information on dataset usage is provided in Appendix C.

**Baselines.** We compare our proposed method against several representative multi-LLM collaboration methods: (i) FrugalGPT (Chen et al., 2024): it evaluates SLM responses with a trained DistillBERT (Sanh et al., 2019) model and routes low-quality queries to the LLM; (ii) HybridLLM (Ding et al., 2024): it employs a trained DeBERTa (He et al., 2021) model to estimate the relative quality of SLM vs. LLM outputs and selects accordingly with cost taken into account; (iii) RouteLLM (Ong et al.): it uses DeBERTa as well and introduces Matrix Factorization (MF) as binary classifiers to select between SLM and LLM. (iv) Router-Bench (Hu et al., 2024): it encodes queries using Sentence Transformer (Reimers & Gurevych, 2019) and later predicts all available LLMs' response quality with KNN and MLP models, followed by cost-aware selection. (v) IRT-Router (Song et al., 2025): it models the routing decision across multiple LLMs using Item Response Theory, enabling effective and interpretable model selection. Given their re-

---

[1] In our experiments, this value is estimated using the average number of output tokens of the remote LLM responses to queries in the training set, which remains consistent among all baselines in Section 4 that need this estimation as well.

spective design scopes, we compare against all of them in the single-remote setting, and against (iv) and (v) in the multi-remote setting.

## 4.2. Main Results

We first compare the proposed method with baselines in terms of the *Cost-Performance AUC* (defined in Eq. 5) in the single-remote scenario across two local SLMs and three datasets. Due to space constraints, we report in the main text only the results obtained with GPT-4o as the remote LLM, while deferring the corresponding results for the other three remote LLMs to Appendix F.5.

The results are presented in Table 1. As shown, the proposed SDR consistently outperforms all baseline methods across different datasets and local SLMs. This indicates that our approach achieves superior overall task performance under varying user budget constraints, while demonstrating strong generalization across diverse local SLM architectures and task domains.

We further observe that the relative performance of different baselines varies across task types. On reasoning-intensive benchmarks such as MMLU and GSM8K, FrugalGPT performs comparatively better, as it requires the SLM to produce an initial response whose intermediate information is critical for the downstream decision module. In contrast, on SQuAD, where the input context already provides sufficient information for reading comprehension, immediate routing methods that do not rely on additional SLM-generated answers achieve competitive performance; moreover, by explicitly predicting the performance of all available models, these methods implicitly capture the performance gap between remote LLMs and local SLMs, whereas cascade-based approaches like FrugalGPT primarily model the absolute performance of the local SLM, which places them at a relative disadvantage in this setting. Motivated by these complementary strengths and limitations, our method jointly considers system workflow design and training strategy, enabling more consistent performance across diverse task types.

Overall, our method is motivated by a careful analysis of the complementary strengths and limitations of these two classes of approaches. By incorporating these insights into both the system workflow and the training strategy, our method achieves more consistent and robust performance across diverse task types.

## 4.3. Multi-remote Scenario

In this setting, we involve all LLMs mentioned in Section 4.1. The comparative results between the proposed Selective Deferred Routing (SDR) and the applicable baseline methods are presented in Figure 4. Due to space con-

straints, we report in the main text only the results obtained with Llama-3.1-8B as the local SLM, while deferring the corresponding results for Qwen-2.5-7B to Appendix F.5.

Since the cost-performance AUC for the single-remote scenario is no longer applicable, we report the curves formed by the actual monetary cost and original performance achieved by different methods. The monetary cost is computed based on the pricing schemes of different LLMs (detailed in Appendix C) and the number of input and output tokens. Data points are selected as follows: we first vary each method's trade-off parameter to identify the points yielding the lowest cost and the highest performance, thereby determining the overall trade-off range. We then sample intermediate points at a specified rate (e.g., 5 intervals in the figure), ensuring that the resulting costs are approximately evenly distributed within this range.

As illustrated in the figure, across all datasets considered, Selective Deferred Routing consistently surpasses the baseline methods at nearly every sampled point, indicating that it delivers strong effectiveness under diverse preferences in the cost–performance trade-off. These results demonstrate that our extension from the single-remote to the multi-remote setting remains effective.

## 4.4. Ablation Study

Relative to existing baselines, SDR simultaneously changes the system workflow, the routing input, and the training objective. To isolate the contribution of each design choice, we conduct controlled ablations that fix the deferred workflow and the performance-gap training objective of SDR, and vary only (i) the routing *input* and (ii) its *representation*.

**Setup.** We consider all combinations over two orthogonal factors. The routing input is either the user prompt alone (**P**) or the prompt concatenated with the local SLM's answer (**P+A**). The representation is either BERT-base text embeddings (**T**) computed by an external encoder, or the local SLM's own hidden states (**H**), as used by SDR. The four resulting variants are denoted SDR (P, T), SDR (P+A, T), SDR (P, H), and SDR (P+A, H). For a fair head-to-head comparison across the four variants, we adopt an MLP decider for all of them in this ablation, since BERT-style text embeddings are not naturally compatible with the transformer-based decider used in the full SDR. All other settings follow the single-remote configuration of Section 4.1.

**Results.** Table 2 reports the cost-performance AUC of the four variants, from which two consistent trends emerge. First, switching the representation from BERT text embeddings to the SLM's own hidden states produces large and consistent gains regardless of the input choice, confirming that the representations produced as a byproduct of SLM

*Table 1.* Comparison of *cost-performance AUC* among proposed method and baselines under different local SLMs and datasets. The remote LLM is GPT-4o. Best results are highlighted in bold, and second-best results are underlined.

| Method | Local SLM: Llama-3.1-8B | | | Local SLM: Qwen-2.5-7B | | |
|---|---|---|---|---|---|---|
| | **MMLU** | **SQuAD** | **GSM8K** | **MMLU** | **SQuAD** | **GSM8K** |
| FRUGALGPT | 0.6465 | 0.4780 | 0.6480 | 0.6366 | 0.4877 | 0.7516 |
| HYBRIDLLM | 0.5930 | 0.5223 | 0.4956 | 0.6082 | 0.5775 | 0.4315 |
| ROUTELLM | 0.6022 | 0.5769 | 0.5141 | 0.6054 | 0.6563 | 0.5101 |
| ROUTERBENCH-KNN | 0.6015 | 0.6307 | 0.5085 | 0.5791 | 0.6366 | 0.6174 |
| ROUTERBENCH-MLP | 0.5445 | 0.6271 | 0.4981 | 0.5646 | 0.6599 | 0.4938 |
| IRT-ROUTER | 0.6171 | 0.5869 | 0.5199 | 0.6112 | 0.5970 | 0.4940 |
| SDR (OURS) | **0.7027** | **0.9165** | **0.6999** | **0.6900** | **0.9225** | **0.7993** |

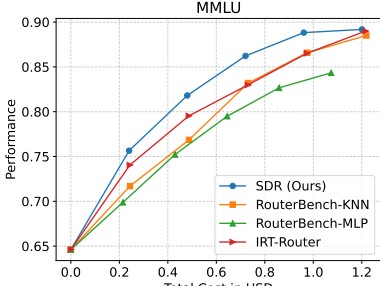 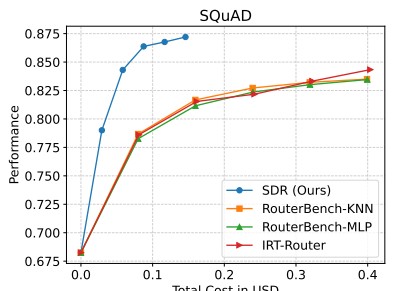 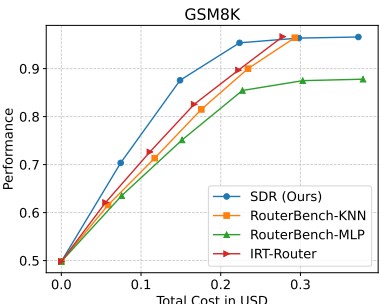

*Figure 4.* Cost-Performance curves of real monetary costs (in USD) and original performance obtained from proposed Selective Deferred Routing (SDR) and baseline methods on 3 datasets in Multi-Remote Scenario. The local SLM is Llama-3.1-8B.

inference are substantially more informative for routing than encoding the same text with an external encoder. Second, conditioning on the SLM answer in addition to the prompt consistently helps under text embeddings, but its marginal benefit shrinks markedly once hidden states are already used. We attribute this to the fact that the hidden states already encode much of the semantic content that the surface-level answer would otherwise expose, so the added value of the textual answer becomes task-dependent and is most visible on reasoning or arithmetic tasks such as GSM8K, where the answer carries strong task-specific signals. Overall, the dominant source of SDR's improvement is the use of the SLM's hidden states as the routing signal, while augmenting the input with the SLM's textual answer provides a complementary, task-dependent boost.

### 4.5. End-to-end Latency Analysis

In this subsection, we analyze the end-to-end latency of a local–remote collaborative system that incorporates our method and baselines. End-to-end latency serves as a critical metric for characterizing the overall user experience. We model the latency of completing a user request on an LLM using the following canonical formulation:

$$\text{latency} = \text{TTFT} + \frac{\text{number of output tokens}}{\text{decoding speed}}. \quad (11)$$

Here, Time-To-First-Token (TTFT) denotes the elapsed time between a user submitting a request and receiving the first output token, which is determined by the prefill time of the serving system and, in the case of remote LLMs, additional network and scheduling delays. The decoding speed refers to the number of tokens that the serving system can decode per unit time.

We first provide a brief qualitative analysis. The local system features a shorter TTFT but a lower decoding speed than the remote system. For tasks requiring long outputs, the TTFT advantage is gradually offset during decoding, allowing the remote system to match or surpass the local system in end-to-end latency; in such cases, local–remote collaboration improves cost efficiency at the expense of higher latency. We provide a brief discussion on potential approaches for further reducing system latency in Appendix E.

By contrast, for tasks with short outputs, the TTFT advantage dominates: a locally served request may complete before a remote request is even scheduled. In this regime, local–remote collaboration is not only cost-efficient but can also yield reduced end-to-end latency as a byproduct.

We next simulate quantitative results on the SQuAD dataset, a reading comprehension benchmark that does not require complex reasoning, resulting in a limited number of output tokens. We set the local system parameters to a TTFT of 0.05 s and a decoding speed of 60 tokens/s, while the

*Table 2.* Controlled ablation isolating the contribution of the routing input and its representation. **P** = prompt only; **P$^{+A}$** = prompt + SLM answer; **T** = BERT-base text embeddings; **H** = SLM hidden states. All four variants use an MLP decider for fair comparison. Best result per column is in bold. The remote LLM is GPT-4o.

| Variant | Local SLM: Llama-3.1-8B | | | Local SLM: Qwen-2.5-7B | | |
|---|---|---|---|---|---|---|
| | MMLU | SQuAD | GSM8K | MMLU | SQuAD | GSM8K |
| SDR (P, T) | 0.6047 | 0.6013 | 0.5476 | 0.6137 | 0.6419 | 0.4975 |
| SDR (P$^{+A}$, T) | 0.6241 | 0.6590 | 0.6113 | 0.6469 | 0.7866 | 0.7123 |
| SDR (P, H) | 0.6509 | **0.8978** | 0.6706 | 0.6684 | 0.9204 | 0.7461 |
| SDR (P$^{+A}$, H) | **0.7008** | 0.8946 | **0.6719** | **0.6695** | **0.9207** | **0.7746** |

*Table 3.* Comparison of average end-to-end latency per request (in seconds) when achieving different levels of performance gain. Best results are highlighted in bold.

| METHOD | 80% PG | 90% PG | 95% PG |
|---|---|---|---|
| FRUGALGPT | 0.9768 | 1.0822 | 1.1353 |
| HYBRIDLLM | 0.8901 | 0.9721 | 0.9992 |
| ROUTERLLM | 0.9713 | 1.0088 | 1.0229 |
| ROUTERBENCH-KNN | 0.7246 | 0.8501 | 0.9116 |
| ROUTERBENCH-MLP | 0.7148 | 0.8532 | 0.9325 |
| IRT-ROUTER | 0.7348 | 0.8502 | 0.9138 |
| SDR (OURS) | **0.3971** | **0.4446** | **0.4710** |
| LOCAL ONLY | | 0.1464 | |
| REMOTE ONLY | | 1.0346 | |

remote system is characterized by a TTFT of 1 s and a decoding speed of 150 tokens/s. The local system metrics are obtained from our own measurements by deploying the local SLMs with vLLM (Kwon et al., 2023) on a single NVIDIA RTX 4090 GPU under standard serving configurations. For remote systems, we choose these values by referencing latency measurements reported in prior studies (Agrawal et al., 2024; Sun et al., 2025) and publicly available third-party benchmarks (Artificial Analysis) of mainstream commercial LLM services. We acknowledge that a single remote-system configuration cannot represent the full diversity of real-world service providers, and thus our chosen parameters correspond to a relatively high-quality remote setting. In practice, many commercial services exhibit TTFTs of several to tens of seconds, whereas the assumed 1 s TTFT is already comparable to typical user–datacenter round-trip latency, making our evaluation a conservative and remote-favoring setting.

The results are reported in Table 3. In this experiment, Qwen-2.5-7B is used as the local SLM and GPT-4o as the remote LLM. For each method, we present the average end-to-end latency per request (in seconds) when achieving different levels of performance gain (defined in Eq. 4). In this setting, the local system exhibits a clear latency advantage over the remote system, making it latency-efficient to handle as many requests locally as possible. This behavior is inherently aligned with the cost-efficient optimization objective introduced in Section 3.2, which in turn leads to superior performance compared to all baselines. Concretely, our method is able to recover 95% of the performance gap between the local SLM and the remote LLM with less than half of the end-to-end latency.

## 5. Conclusion

In this work, we introduced Selective Deferred Routing (SDR), a cost-efficient paradigm for collaborative inference between local SLMs and remote LLMs. By first leveraging a local SLM to generate both a preliminary response and informative representations, our approach enables a lightweight decision module to make informed routing decisions without incurring unnecessary remote invocation costs. Extensive experiments across diverse model families, architectures, and task scenarios demonstrate that our method consistently achieves superior cost–performance trade-offs compared to existing multi-LLM collaboration strategies. These results highlight the effectiveness and robustness of selectively deferring computation to remote LLMs, and suggest a practical path toward scalable and economical LLM-powered systems.

There are also many promising directions for future work. For instance, post-training local SLMs to enhance their self-evaluation capability or to produce more concise outputs may further improve the system's cost efficiency. In addition, improving the smoothness of decision scores may yield more stable cost-performance behavior under varying thresholds, better matching user preferences.

## Acknowledgements

The authors thank the anonymous reviewers for their helpful comments. This work was supported by Xiongan AI Institute and Tsinghua University Dushi Program.

## Impact Statement

This work proposes Selective Deferred Routing (SDR), a cost-efficient collaboration paradigm between local SLMs and remote LLMs. By enabling user requests to be selec-

tively handled by local models and routed to remote models only when necessary, the proposed approach has the potential to reduce monetary costs, network usage, and energy consumption. This may lower the barrier to accessing high-quality language model services for individuals and small organizations with limited computational or financial resources, and support more sustainable deployment of LLM-based systems.

At the same time, the deployment of routing mechanisms between local and remote models introduces potential risks. For example, excessive reliance on local models could propagate their biases or limitations.

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

## A. Proof of the Optimal BSR Theorem

According to the definition of Cost in Eq. 2, for a finite dataset $\mathcal{Q} = \{q_i\}_{i=1}^N$, the possible values of Cost are restricted to $\{\frac{i}{N} \mid i \in [0, N]\}$.

For all $q_i \in \mathcal{Q}$, we sort the elements in ascending order according to their scores given by the scoring model $s_i$, resulting in the sequence $(q_{(1)}, q_{(2)}, \cdots, q_{(N)})$, with the corresponding ascending scores $(s_{(1)}, s_{(2)}, \cdots, s_{(N)})$. We can then establish a one-to-one correspondence between the range of the decision threshold $\alpha$ and the resulting Cost:

$$\begin{cases} \alpha \in (-\infty, s_{(1)}] \Leftrightarrow \text{Cost}(\alpha) = 0, \\ \alpha \in (s_{(i)}, s_{(i+1)}] \Leftrightarrow \text{Cost}(\alpha) = \frac{i}{N}, \forall i \in [1, N), \\ \alpha \in (s_{(N)}), +\infty) \Leftrightarrow \text{Cost}(\alpha) = 1. \end{cases}$$

Therefore, we can select a representative $\alpha$ from each except the first[2] of the above intervals to form a sequence $(\alpha_1, \cdots, \alpha_N)$ to reformulate the AUC in Eq. 5 as follows:

$$\text{AUC} = \frac{1}{N} \sum_{i=1}^{N} \text{PG}(\alpha_i).$$

According to the definitions in Eq. 4 and Eq. 6, the performance gain can be computed as the prefix sum of the performance gaps along the sequence of $q$ sorted as described above:

$$\text{PG}(\alpha_i) = \frac{\sum_{j=1}^{i} \text{Gap}_{(j)}}{\lim_{\alpha_1 \to +\infty} \text{Perf}(\alpha_1) - \lim_{\alpha_0 \to -\infty} \text{Perf}(\alpha_0)} = \frac{\sum_{j=1}^{i} \text{Gap}_{(j)}}{\sum_{j=1}^{N} \text{Gap}_{(j)}}.$$

Consequently, the cost-performance AUC can be further rewritten as:

$$\text{AUC} = \frac{1}{N \sum_{j=1}^{N} \text{Gap}_{(j)}} \sum_{i=1}^{N} \sum_{j=1}^{i} \text{Gap}_{(j)}.$$

From the form of the above expression, it follows directly from the rearrangement inequality that the sum of prefix sums is maximized when the array is sorted in non-increasing order. Hence, Theorem 3.1 is established.

## B. Budget-Aware Threshold Calibration

While AUC provides a threshold-independent evaluation of routing quality, practical deployment requires selecting a concrete operating threshold under user-specified cost constraints. Following the discussion in Dekoninck et al. (2025), such threshold selection can be conducted as a budget-constrained performance optimization using a validation dataset.

Specifically, we first construct a validation dataset that is representative of the target query distribution. Given a trained SDR decision module, we compute a routing score for each validation sample. We then perform a threshold search to identify the operating point that maximizes overall task performance while satisfying a predefined average cost budget.

In practice, it's worth noting that inference cost and performance generally exhibit a monotonic relationship with respect to the routing threshold. Consequently, the feasible threshold can be efficiently identified via binary search instead of exhaustive sweeping. This procedure provides a practical pipeline for translating user-level budget requirements into concrete deployment configurations.

## C. Additional Details of the Experiment Settings

**LLMs Using.** For local SLMs, we deploy their instruction-tuned versions locally and apply the default chat templates together with greedy decoding to generate responses on the datasets. For the remote LLMs, we obtain responses on the datasets via cloud service providers, using sampling parameters set to temperature = 1, top-p = 1, and top-k disabled.

---

[2]We do not need to select an $\alpha$ from the interval corresponding to $\text{Cost}(\alpha) = 0$, as this would not yield any performance gain.

For prompt organization, we adapt to the nature of each dataset: in MMLU and GSM8K, we instruct the LLMs to produce a concise reasoning process followed by the final answer. Additionally, we require remote LLMs' final answers to follow a fixed output format, enabling direct answer extraction. In contrast, for local SLMs, we do not impose strict formatting constraints on the final answers, as their limited ability to adhere to structured output instructions could otherwise negatively affect their true task performance. Instead, we use a more capable LLM (i.e., o4-mini) to extract the final answers from their free-form responses. In SQuAD, we require all the LLMs to output only the final answers. The full prompts used for answer generation and final answer extraction are shown below.

Generating answers for MMLU dataset:

```
You are given a multiple choice problem. Provide a brief reasoning then a
final answer.

Question: {question}
A: {option A}
B: {option B}
C: {option C}
D: {option D}
```

Generating answers for SQuAD dataset:

```
You are given a reading comprehension problem. Response with **only** your
answer (in a short phrase).

Context: {context}

Question: {question}
```

Generating answers for GSM8K dataset:

```
You are given a math problem. Provide a brief reasoning then a final answer.

Question: {question}
```

Extracting the final answer for MMLU and GSM8K datasets:

```
You are an impartial answer extractor for evaluating a language model.

You will be given:
1. A question from a dataset
2. A free-form response generated by a model

Your task is to extract and normalize the model's final answer, without
judging correctness.

General rules:
- Do NOT judge correctness.
- Do NOT use external knowledge.
```

```
- Do NOT infer missing information or complete incomplete answers.
- Only extract what the model explicitly commits to as its final answer.
- If no definite final answer can be identified, output "UNANSWERABLE".

Task-specific rules:

1. If the question is a single-choice (multiple-choice with one correct
option) question:
- The final answer MUST be the letter corresponding to the chosen option (e.
g., A, B, C, D).
- If the model response explicitly states an option letter, extract that
letter.
- If the model response only states the content of an option (but not the
letter), map it to the corresponding option letter using the given question.
- If multiple options are mentioned, or the choice cannot be uniquely
determined, output "UNANSWERABLE".

2. If the question is a mathematics question:
- The final answer MUST be a single number, either an integer or a decimal.
- Extract the final numerical value explicitly stated by the model.
- Do NOT perform any calculation, simplification, rounding, or unit
conversion.
- If multiple numbers are given and no unique final answer is clearly
indicated, output "UNANSWERABLE".

Output format:
Return ONLY the final answer, with no explanation, no punctuation, and no
additional text.
```

For evaluation, correctness on MMLU and GSM8K is determined by exact matching with the reference answers. For SQuAD, we follow the official recommendation and use the maximum F1 score between the model's output and the multiple reference answers as the measure of answer quality. The resulting performances for each LLM on datasets are shown in Table 4.

*Table 4.* Local SLM and remote LLMs' average performances on test splits of 3 datasets (normalized to 1).

| Model | MMLU | SQuAD | GSM8K |
|---|---|---|---|
| LLAMA-3.1-8B | 0.6461 | 0.6825 | 0.4981 |
| QWEN-2.5-7B | 0.7124 | 0.6086 | 0.9045 |
| LLAMA-4-MAVERICK | 0.8893 | 0.8202 | 0.9666 |
| DEEPSEEK-V3 | 0.8861 | 0.8429 | 0.9644 |
| O4-MINI | 0.8911 | 0.7930 | 0.9613 |
| GPT-4O | 0.8747 | 0.8214 | 0.9429 |

**Datasets Using.** We use the test split of the MMLU (Hendrycks et al., 2021) and the validation splits of the SQuAD (Rajpurkar et al., 2016), further dividing each into training and test sets with an 8:2 ratio, since their training splits are excessively large. For GSM8K (Cobbe et al., 2021), we directly adopt its official training and test splits.

**LLMs Pricing.** We assume that the local SLM does not incur any monetary cost. The pricing data of the remote LLMs used in our experiments are summarized in Table 5. For the open-source models Llama-4-Maverick (Meta AI, 2025), and DeepSeek-V3 (DeepSeek-AI, 2024), we adopt the average pricing across all API providers listed on Artificial Analysis. For the proprietary models (o4-mini and GPT-4o), we follow their official API pricing (OpenAI).

*Table 5.* LLMs pricing data used in experiments.

| Model | Input (USD per million Tokens) | Output (USD per million Tokens) |
|---|---|---|
| Llama-4-Maverick | 0.29 | 0.93 |
| DeepSeek-V3 | 1.00 | 1.54 |
| o4-mini | 1.10 | 4.40 |
| GPT-4o | 2.50 | 10.00 |

*Table 6.* Comparison of computational and memory overhead between the vanilla SLMs and our proposed SDR integrated model.

| METRIC | SLM ONLY | SLM + DECISION MODULE | OVERHEAD ($\Delta$) | RATIO |
|---|---|---|---|---|
| LOCAL SLM: LLAMA-3.1-8B | | | | |
| PARAMETERS | 8.03 B | 8.25 B | +0.22 B | +2.78% |
| LATENCY (PREFILL) | 56.62 MS | 58.85 MS | +2.24 MS | +3.95% |
| LATENCY (DECODE) | 15.16 MS/TOKEN | 15.75 MS/TOKEN | +0.59 MS/TOKEN | +3.40% |
| PEAK VRAM | 15.16 GB | 15.75 GB | +0.59 GB | +3.89% |
| LOCAL SLM: QWEN-2.5-7B | | | | |
| PARAMETERS | 7.62 B | 7.85 B | +0.23 B | +3.12% |
| LATENCY (PREFILL) | 54.74 MS | 57.18 MS | +2.44 MS | +4.45% |
| LATENCY (DECODE) | 27.83 MS/TOKEN | 29.01 MS/TOKEN | +1.18 MS/TOKEN | +4.26% |
| PEAK VRAM | 14.41 GB | 15.01 GB | +0.60 GB | +4.17% |

## D. Latency and Memory Overhead Analysis of Decision Module

We evaluate the computational and memory overhead introduced by integrating the proposed Selective Deferred Routing (SDR) framework into local SLM inference. All experiments are conducted on a single NVIDIA RTX 4090 GPU with a straightforward implementation using PyTorch. To measure prefill latency, we use a fixed prompt length of 512 tokens and report the average latency over 100 independent runs. For decoding latency, the same 512-token prompt is used, and the model is instructed to generate 32 new tokens; we report the average per-token decoding latency. The peak GPU memory usage is measured under the same prefill setting, as memory consumption during decoding increases progressively with the number of generated tokens due to the accumulation of KV cache.

Table 6 compares the parameter count, inference latency, and peak VRAM usage between the vanilla SLMs and the SDR-integrated models for both Llama-3.1-8B and Qwen-2.5-7B. As shown, the additional overhead introduced by the decision module remains modest across all metrics. In particular, the parameter increase is below 3.2%, while both prefill and decoding latency incur less than 5% relative overhead. Similarly, the peak VRAM usage increases by approximately 4%, indicating that the proposed framework imposes only a limited memory footprint.

These results highlight the benefits of our lightweight decision module design, which avoids imposing substantial computational or memory burdens on local inference. It is worth noting that all measurements are obtained using a straightforward implementation without specialized optimization. Notably, the architecture of the decision module is designed to enable partial parallel execution with the local SLM, suggesting that further implementation-level optimizations could further reduce the observed overhead in practical deployments. Exploring such system-level optimizations represents a promising direction for future work.

## E. Discussion on Optimizations of the End-to-end Latency

As discussed in Section 4.5, for user requests that tend to elicit long outputs from the local SLM, the relatively slower decoding speed of the local system compared to remote LLM services can lead to higher end-to-end latency than directly routing the request to a remote LLM. In this section, we discuss several potential approaches to mitigate this issue, with the goal of providing insights for future work.

A straightforward and easy-to-implement approach is to impose a limitation on the maximum number of tokens generated

by the local SLM. Once this limit is reached, the system can trigger the selective routing decision without waiting for the SLM to produce an end-of-sequence token. However, this strategy entails several potential drawbacks. First, contemporary LLMs are often encouraged to perform extensive reasoning to enhance their problem-solving capabilities, which may result in intermediate outputs that differ semantically, or even contradict, the final response (Kumar et al., 2025; Kamoi et al., 2024; Zhang et al., 2025). A typical example arises during chain-of-thought reasoning, where a model may revise its answer after producing intermediate statements such as "wait, let's double-check the answer." Truncating the output prior to such revisions may yield a context that does not faithfully reflect the local SLM's true capability. This can lead to unnecessary routing of queries that the local SLM would otherwise answer correctly, and may also adversely affect the performance of the decision module trained on complete SLM responses. Second, the appropriate truncation length cannot be predetermined in a universal manner, as different user queries inherently require different token budgets to support meaningful reasoning.

Another approach is to perform post-training on the local SLM to encourage more concise and well-structured outputs, which has been well explored in prior work (Liu et al., 2024; Cui et al., 2025; Ma et al., 2025). This strategy is largely orthogonal to our routing method and, in principle, does not interfere with the effectiveness of the decision module, making it a more promising direction for reducing latency.

In summary, to further optimize the end-to-end latency of the proposed local–remote collaboration system, we view post-training the local SLM to better regulate its generation behavior, potentially in combination with adaptive maximum output token constraints, as a promising avenue for future research.

## F. Supplementary Experiment Results

### F.1. Alternative Learning-to-Rank Objectives

As stated in Theorem 3.1, the local–remote collaboration problem fundamentally requires the predicted scores produced by the decision module to be aligned with the performance gap between the remote LLMs and the local SLM. In the main body of this paper, we adopt the simplest point-wise formulation and train the decision module using a mean squared error (MSE) loss. This choice is motivated by its simplicity, stability, and strong empirical performance. Moreover, this training objective endows the predicted scores with a clear numerical interpretation, which allows them to be seamlessly incorporated into the routing decisions in the multi-remote scenario described in Section 3.4. In this section, we briefly review several representative learning-to-rank objectives and report our experimental results when applying these alternatives to our setting.

Learning-to-rank methods (Cao et al., 2007) are commonly categorized into three classes depending on how training supervision is defined and how ranking relationships are modeled: point-wise, pair-wise, and list-wise.

*Point-wise* methods treat the ranking problem as a standard regression or classification task, where each candidate item is independently assigned a relevance score. In our context, this corresponds to directly regressing the estimated performance gap for each remote LLM. Point-wise objectives are easy to implement, computationally efficient, and compatible with standard optimization pipelines. However, they do not explicitly model relative ordering among candidates and therefore may be suboptimal when the primary objective is to preserve ranking consistency rather than absolute score accuracy.

*Pair-wise* methods formulate ranking as a comparison problem between pairs of candidates, encouraging the model to assign higher scores to better-performing options within each pair. By directly optimizing relative preferences, pair-wise losses are more aligned with ranking quality and are often more robust to scale variations in the target scores. That said, their training cost grows with the number of candidate pairs, and they may introduce additional variance due to sampling strategies, especially when the number of candidates is limited.

*List-wise* methods optimize ranking quality at the level of an entire candidate list, explicitly modeling interdependencies among all candidates within the same list. Such objectives are well suited for scenarios where each training instance is associated with a small set of candidates and the goal is to produce a relative ordering within that set. However, in our setting, the objective is to align predicted scores with the performance gap between local and remote models at a global, dataset-wide level, rather than to rank a small number of candidates within each instance, making list-wise formulations are not well aligned with our problem and are therefore less applicable in practice.

In addition to the standard point-wise MSE loss used in the main text, we also experiment with a pair-wise ranking objective proposed in RankNet (Burges et al., 2005) and a hybrid training objectives that equally combine point-wise regression with pair-wise ranking losses (Sculley, 2010).

*Table 7.* Comparison of *cost-performance AUC* obtained from three training objectives under different local SLMs and datasets. The remote LLM is GPT-4o.

| Training Objective | Local SLM: Llama-3.1-8B | | | Local SLM: Qwen-2.5-7B | | |
|---|---|---|---|---|---|---|
| | MMLU | SQuAD | GSM8K | MMLU | SQuAD | GSM8K |
| POINT-WISE (MSE) | 0.7027 | 0.9165 | 0.6999 | 0.6900 | 0.9225 | 0.7993 |
| PAIR-WISE (RANKNET) | 0.7086 | 0.8950 | 0.6997 | 0.6804 | 0.9144 | 0.5845 |
| HYBRID | 0.7012 | 0.9193 | 0.6990 | 0.6865 | 0.9264 | 0.7016 |

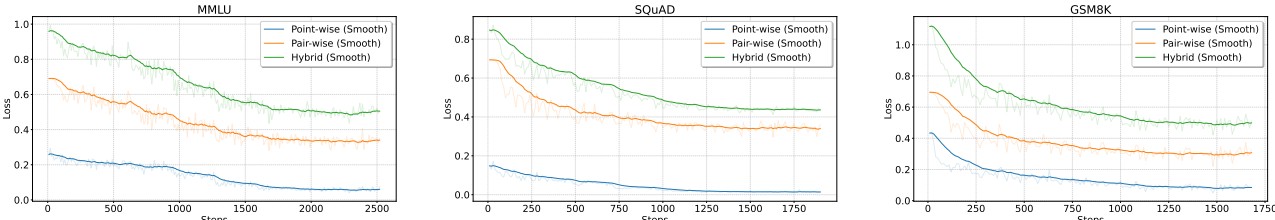

*Figure 5.* Training loss curves of different objectives on three datasets. The local SLM is Llama-3.1-8B.

As shown in the Table 7, while these objectives exhibit slight performance variations across different datasets, no single training objective consistently dominates the others. The observed differences in cost–performance AUC are marginal, indicating that the overall system performance is relatively insensitive to the specific choice of ranking loss in our setting.

In addition, we report the training dynamics of different objectives by visualizing their loss curves in Figure 5 and Figure 6. The point-wise MSE objective demonstrates faster convergence and more stable loss values throughout training, whereas pair-wise and hybrid objectives tend to exhibit higher variance and slower convergence. This observation is consistent with prior findings in the learning-to-rank literature (Cao et al., 2007), which suggest that point-wise objectives often provide more stable optimization behavior, particularly when the target scores are required to retain a clear numerical interpretation.

### F.2. Alternative Designs of the Decision Module

We first explore different inputs to the decision module. As shown in Figure 7 and Figure 8, we report the cost–performance AUC achieved on three datasets when using hidden states extracted from different layers of the local SLM as decision features. Specifically, we consider the last 1/2, 1/3, and 1/4 of layers, as well as the penultimate and final layers, with the local SLM instantiated as Llama-3.1-8B and Qwen-2.5-7B across three datasets. The remote LLM is instantiated as GPT-4o.

While the optimal choice varies across datasets and model backbones, a consistent trend emerges: using the final-layer hidden state is generally suboptimal for the downstream decision task, with the performance degradation being particularly pronounced for Qwen-2.5-7B. This observation aligns with prior studies showing that deeper-layer representations, which are more strongly aligned with the pretraining or instruction-tuning objectives, tend to exhibit weaker generalization to downstream tasks (Van Aken et al., 2019; Skean et al., 2024; 2025). We note that another line of work has explored simple or adaptive aggregation of hidden states from multiple layers to construct higher-quality features (Boquio & Naval Jr, 2024; Skean et al., 2025); however, as this direction falls outside the main focus of this work, we leave a systematic investigation to future studies.

We also evaluate alternative architectural choices for the decision module. As illustrated in Figure 9, we report the cost–performance AUC on three datasets when using Llama-3.1-8B and Qwen-2.5-7B as the local SLM, with GPT-4o serving as the remote LLM, under three different decision module designs: a single linear layer, an MLP, and a transformer block. The results show that the transformer-based design achieves the best performance, as it is able to most effectively exploit the per-token information produced by the local SLM. Notably, even the simple MLP attains strong performance and consistently outperforms all baseline methods reported in Section 4, further highlighting the rich and informative semantic representations encoded in the hidden states of the local SLM.

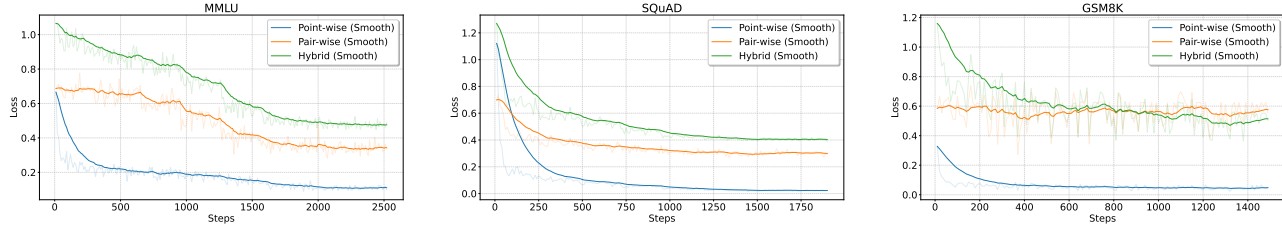

*Figure 6.* Training loss curves of different objectives on three datasets. The local SLM is Qwen-2.5-7B.

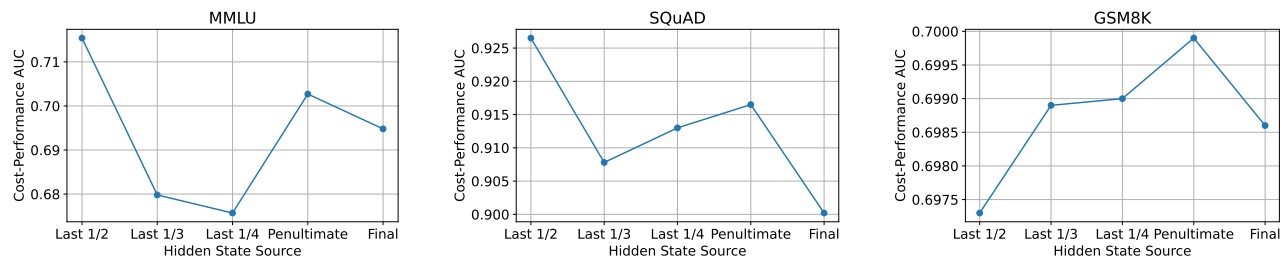

*Figure 7.* Cost–performance AUC achieved on three datasets when using hidden states extracted from different layers of the local SLM (Llama-3.1-8B). The remote LLM is GPT-4o.

### F.3. Different Designs to Adapt in Multi-remote Scenario

We first recap that, in the single-remote scenario, the decision module consists of a transformer layer followed by an MLP. For the multi-remote scenario, we explore three variants of the decision module with different degrees of parameter sharing across remote LLMs: (i) Fully Independent, where each remote LLM is assigned a dedicated decision module; (ii) Fully Shared, where all remote LLMs share a single decision module; and (iii) Shared Transformer + Independent MLPs, where a transformer layer is shared across all remote LLMs, followed by a separate MLP head for each remote LLM.

We compare the resulting cost–performance curves of these three designs across three datasets, as shown in Figures 10 and 11.

We observe that the fully shared scheme exhibits the most stable performance across all settings, whereas the other two schemes suffer from noticeable performance degradation under certain conditions. We attribute this phenomenon to two potential factors. First, decision making among multiple remote LLMs inherently requires the scorer to capture their relative performance differences for a given request, a property that is naturally facilitated by shared parameters. Second, as the degree of parameter sharing decreases, the total number of introduced parameters increases accordingly. Given the moderate size of our datasets (on the order of 10k samples), this design is more susceptible to overfitting, which may further impair generalization performance.

### F.4. Generalization Analysis

Beyond the standard single-dataset evaluation setting, we further evaluate our method under two more challenging scenarios designed to better reflect practical deployment conditions.

First, we construct a *Mixed* setting by combining samples from MMLU, SQuAD, and GSM8K into a unified evaluation benchmark. This setting better approximates realistic user workloads, where incoming queries may span multiple domains and task types simultaneously. The train-test partitions remain consistent with those of the original datasets.

Second, we introduce an out-of-distribution (*OOD*) setting to evaluate cross-dataset generalization. Concretely, the decision module is trained on the SQuAD training set and evaluated on CoQA (Reddy et al., 2019). Both datasets belong to the reading comprehension domain and differ in data distribution, making this setting a more challenging test of generalization ability.

As shown in Table 8, our method consistently outperforms all baseline methods under both settings. These results further demonstrate the practical applicability of SDR in realistic multi-domain deployment scenarios, as well as its strong robustness

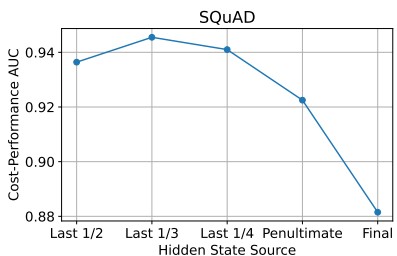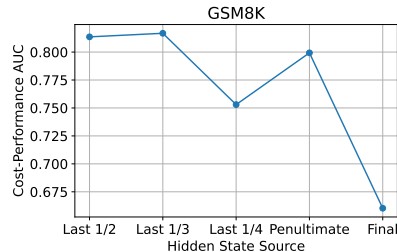

*Figure 8.* Cost–performance AUC achieved on three datasets when using hidden states extracted from different layers of the local SLM (Qwen-2.5-7B). The remote LLM is GPT-4o.

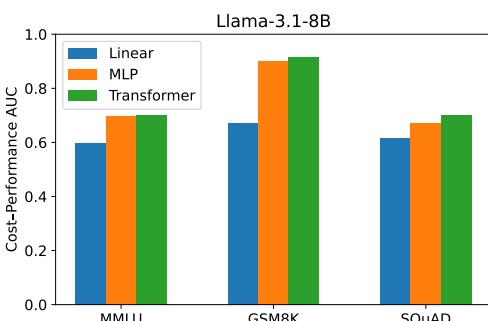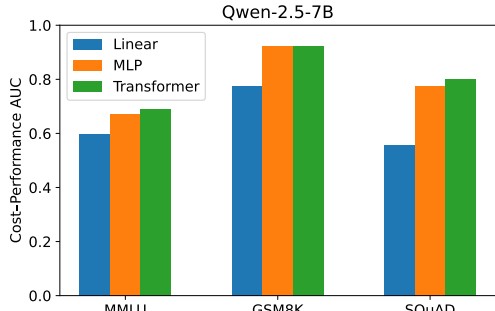

*Figure 9.* Cost–performance AUC under three different decision module designs on three datasets using Llama-3.1-8B (left) and Qwen-2.5-7B (right) as local SLMs. The remote LLM is GPT-4o.

under distribution shifts.

### F.5. Additional Results for Section 4

In this section, we report additional experimental results from Section 4.2 and Section 4.3 that were omitted from the main text due to space limitations.

Tables 9, 10 and 11 present comparisons of cost–performance AUC between our method and baseline approaches on three datasets under the single-remote scenario, where the remote LLM is instantiated as Llama-4, DeepSeek-V3, and o4-mini, respectively. As shown, the proposed SDR consistently outperforms all baseline methods across different datasets and local SLMs, demonstrating its generalizable effectiveness across remote LLMs with varying capability profiles.

Figure 12 further illustrates the trade-off curves between actual monetary cost and original task performance in the multi-remote scenario, with the local SLM set to Qwen-2.5-7B, across three datasets. As shown, our method consistently surpasses the baseline methods at nearly all sampled operating points, indicating that SDR generalizes effectively to multi-remote settings and remains robust across local SLMs with different architectures.

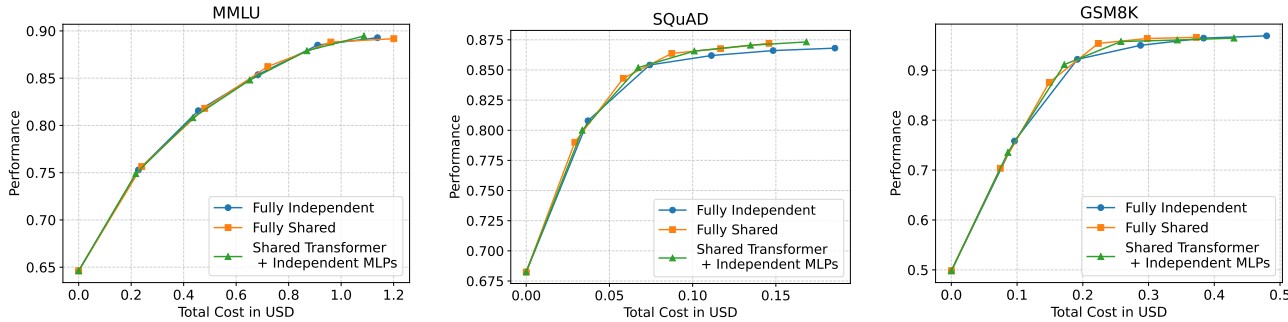

*Figure 10.* Cost-Performance curves of real monetary costs (in USD) and original performance obtained from different decision module designs on 3 datasets in Multi-Remote Scenario. The local SLM is Llama-3.1-8B.

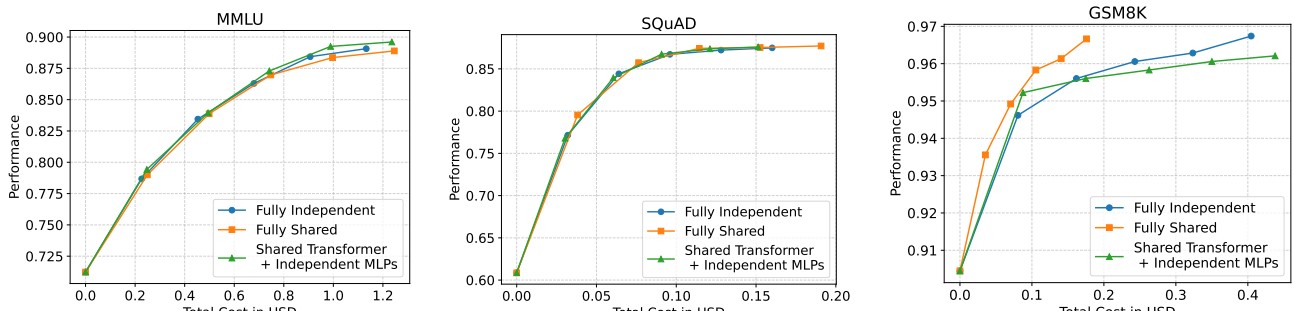

*Figure 11.* Cost-Performance curves of real monetary costs (in USD) and original performance obtained from different decision module designs on 3 datasets in Multi-Remote Scenario. The local SLM is Qwen-2.5-7B.

*Table 8.* Comparison of *cost-performance AUC* among proposed method and baselines under different local SLMs and two additional settings. The remote LLM is GPT-4o. Best results are highlighted in bold, and second-best results are underlined.

| Method | Local SLM: Llama-3.1-8B | | Local SLM: Qwen-2.5-7B | |
|---|---|---|---|---|
| | MIXED | OOD | MIXED | OOD |
| FRUGALGPT | 0.6709 | 0.5126 | 0.7780 | 0.5046 |
| HYBRIDLLM | 0.5310 | 0.5087 | 0.6458 | 0.5246 |
| ROUTELLM | 0.4997 | 0.4971 | 0.5090 | 0.5078 |
| ROUTERBENCH-KNN | 0.5019 | 0.5007 | 0.4958 | 0.4972 |
| ROUTERBENCH-MLP | 0.4975 | 0.5016 | 0.5068 | 0.4942 |
| IRT-ROUTER | 0.5097 | 0.5000 | 0.4995 | 0.4991 |
| SDR (OURS) | **0.7765** | **0.5314** | **0.8497** | **0.5658** |

*Table 9.* Comparison of *cost-performance AUC* among proposed method and baselines under different local SLMs and datasets. The remote LLM is Llama-4-Maverick. Best results are highlighted in bold, and second-best results are underlined.

| Method | Local SLM: Llama-3.1-8B | | | Local SLM: Qwen-2.5-7B | | |
|---|---|---|---|---|---|---|
| | MMLU | SQuAD | GSM8K | MMLU | SQuAD | GSM8K |
| FRUGALGPT | 0.6565 | 0.4876 | 0.6480 | 0.6429 | 0.4854 | 0.7149 |
| HYBRIDLLM | 0.6088 | 0.5505 | 0.5286 | 0.6207 | 0.5453 | 0.5493 |
| ROUTELLM | 0.6199 | 0.5669 | 0.5182 | 0.6201 | 0.6149 | 0.5418 |
| ROUTERBENCH-KNN | 0.6003 | 0.5816 | 0.4968 | 0.6082 | 0.5736 | 0.4851 |
| ROUTERBENCH-MLP | 0.5646 | 0.5939 | 0.4956 | 0.5761 | 0.5846 | 0.3863 |
| IRT-ROUTER | 0.6286 | 0.5476 | 0.5144 | 0.6329 | 0.5469 | 0.5033 |
| SDR (OURS) | **0.7153** | **0.9650** | **0.7007** | **0.6952** | **0.9021** | **0.8076** |

*Table 10.* Comparison of *cost-performance AUC* among proposed method and baselines under different local SLMs and datasets. The remote LLM is DeepSeek-V3. Best results are highlighted in bold, and second-best results are underlined.

| Method | Local SLM: Llama-3.1-8B | | | Local SLM: Qwen-2.5-7B | | |
|---|---|---|---|---|---|---|
| | MMLU | SQuAD | GSM8K | MMLU | SQuAD | GSM8K |
| FRUGALGPT | 0.6516 | 0.4834 | 0.6492 | 0.6495 | 0.4930 | 0.7351 |
| HYBRIDLLM | 0.6049 | 0.5751 | 0.4946 | 0.6332 | 0.5758 | 0.5342 |
| ROUTELLM | 0.6172 | 0.5984 | 0.5112 | 0.6152 | 0.6456 | 0.5153 |
| ROUTERBENCH-KNN | 0.5673 | 0.5904 | 0.5011 | 0.5792 | 0.5798 | 0.5468 |
| ROUTERBENCH-MLP | 0.5575 | 0.5871 | 0.4945 | 0.5569 | 0.5935 | 0.4324 |
| IRT-ROUTER | 0.6280 | 0.5873 | 0.5161 | 0.6313 | 0.5878 | 0.5046 |
| SDR (OURS) | **0.7018** | **0.8864** | **0.6991** | **0.6713** | **0.8894** | **0.7928** |

*Table 11.* Comparison of *cost-performance AUC* among proposed method and baselines under different local SLMs and datasets. The remote LLM is o4-mini. Best results are highlighted in bold, and second-best results are underlined.

| Method | Local SLM: Llama-3.1-8B | | | Local SLM: Qwen-2.5-7B | | |
|---|---|---|---|---|---|---|
| | MMLU | SQuAD | GSM8K | MMLU | SQuAD | GSM8K |
| FRUGALGPT | 0.6608 | 0.4750 | 0.6521 | 0.6446 | 0.4881 | 0.7632 |
| HYBRIDLLM | 0.6257 | 0.5312 | 0.5141 | 0.6268 | 0.5824 | 0.5652 |
| ROUTELLM | 0.6249 | 0.5931 | 0.5151 | 0.6275 | 0.6712 | 0.5286 |
| ROUTERBENCH-KNN | 0.5744 | 0.6096 | 0.4963 | 0.6023 | 0.6108 | 0.4735 |
| ROUTERBENCH-MLP | 0.5514 | 0.6193 | 0.4941 | 0.5720 | 0.6075 | 0.4427 |
| IRT-ROUTER | 0.6291 | 0.6000 | 0.5136 | 0.6357 | 0.6074 | 0.5032 |
| SDR (OURS) | **0.7079** | **1.0245** | **0.7032** | **0.7032** | **0.9976** | **0.8270** |

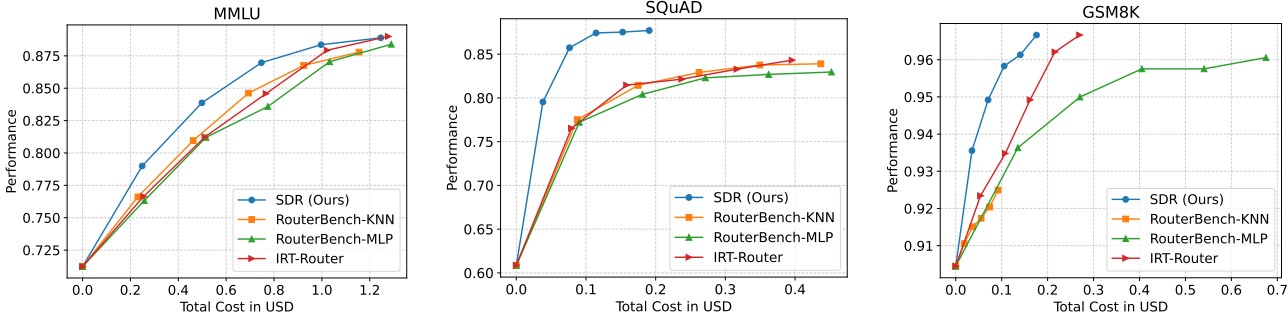

*Figure 12.* Cost-Performance curves of real monetary costs (in USD) and original performance obtained from proposed Selective Deferred Routing (SDR) and baseline methods on 3 datasets in Multi-Remote Scenario. The local SLM is Qwen-2.5-7B.

