# OpenReview forum: "Selective Deferred Routing: Enabling Cost-Efficient Collaboration between Local SLMs and Remote LLMs"
_ICML.cc/2026/Conference — ICML 2026 regular_

### Official Review · Reviewer_qgs4 · 2026-03-03

**Soundness:** 3
**Presentation:** 3
**Significance:** 3
**Originality:** 2
**Overall Recommendation:** 5
**Confidence:** 4

**Summary:**

This paper proposes Selective Deferred Routing (SDR) for cost-efficient collaboration between local SLMs and remote LLMs. The local SLM first processes user requests, generating both a textual response and intermediate hidden states. A lightweight decision module (single transformer block + MLP) then leverages these hidden states to decide whether to accept the local response or route to a remote LLM. The routing objective is formalized to evaluete the qaulity gap rather than the SLM's absolute quality. Experiments on MMLU, SQuAD, and GSM8K with two local SLMs and four remote LLMs demonstrate consistent improvements over six baselines.

**Compliance With Llm Reviewing Policy:**

Affirmed.

**Final Justification:**

I think it's a promising direction of extracting hidden layer as input, together with the original query to feed the router to make decision and their experiment results validate the advantage of their methods.

During the rebuttal, they provided additional ablation study to show the actual effect of each component, which clearly shows the benefit of the hidden representation.

In general, I think this paper can inspire future research on LLM Router/Cascade. I choose to increase my score to 5.

**Key Questions For Authors:**

1. Can you ablate hidden states vs. textual features (prompt + SLM response via BERT/DeBERTa) under matched conditions?
2. How does SDR compare to AutoMix or Dekoninck's method experimentally?

**Limitations:**

1. Requires the access to the internal hidden states of SLM.

2. The router seems to be retrained for every single benchmark. Then the performance of the light-weight router on more diverse tasks or daily queries is unclear.

**Strengths And Weaknesses:**

## Strengths

**S1: Well-motivated design.** The paper outlines a general aspect of the tension between immediate routing (limited decision signals) and model cascades (sequential overhead, rigid ordering assumptions). SDR strikes a practical middle ground: the SLM always runs first, providing enriched features for a one-shot routing decision. This avoids cascading's worst case while utilizing far richer information than prompt-only routing.

**S2: Strong empirical results.** SDR outperforms all baselines across nearly every setting, often by large margins (e.g., ~45% relative improvement on SQuAD). Results hold across both single-remote and multi-remote scenarios and four remote LLMs. The latency analysis shows SDR recovers 95% of the performance gap at less than half the baseline latency on short-output tasks.

**S3: Hidden states for routing is a compelling contribution.** The authors strive to discuss a central concept: exploiting the SLM's internal representations rather than treating it as a black box. Applying hidden-state probing to multi-LLM routing decisions is novel. The scoring model design is practical, adding around 3.2% parameters, around 5% latency, and around 4% memory overhead.


## Weaknesses

**W1: Deferred routing is not a new paradigm, and key related work is missing.**

The "generate first, then decide" workflow already exists in AutoMix (Aggarwal et al., 2023), which the paper cites but misclassifies as a cascade method. AutoMix's structure — SLM generates, evaluates confidence, escalates if insufficient — is fundamentally the same as SDR's workflow. The real difference is in the decision signal (self-evaluation vs. hidden states) and training objective, not the paradigm itself. Moreover, AutoMix is not included as a baseline despite being the most directly comparable method.

Additionally, Dekoninck et al. ("A Unified Approach to Routing and Cascading for LLMs," ICML 2025) propose a unified routing-cascading framework with AUC-based evaluation in a highly related setting. This work is neither cited nor compared against.

**W2: Missing ablations to isolate each component's contribution.**

SDR simultaneously changes the workflow, input features, and training objective relative to baselines. Without controlled ablations, the source of improvement is unclear. The following conditions under the same deferred framework and performance-gap objective are needed:

- (a) Prompt only
- (b) Prompt + SLM textual response
- (c) Hidden states only
- (d) Full information


Figure 9's architecture ablation (Linear/MLP/Transformer) only compares different ways of processing hidden states, not whether they are necessary.



**W3: Unaddressed threshold calibration for practical deployment.**

The routing decision depends on a threshold \alpha (single-remote) or trade-off parameter k (multi-remote). The paper sweeps these for evaluation curves but does not discuss how a user translates a concrete budget or performance constraint into an appropriate \alpha  or k. This mapping depends on the score distribution of incoming queries, which varies across domains. A validation-set calibration procedure and score calibration analysis would strengthen the paper.

---

## Minor Issues

- Typo on page 8: "less thanzhu half."

---

> ### Author Rebuttal · Authors · 2026-03-30
>
> Dear Reviewer qgs4,
>
> We sincerely thank you for your thoughtful review and appreciate your constructive feedback, which we will carefully address in the upcoming revision. Below, we address your specific questions and concerns:
>
> # 1. Related Work Issues (to W1)
> First, we would like to sincerely thank you for pointing out the omissions in our literature review; we will correct these presentation issues in the revised manuscript. We would also like to **clarify the fundamental differences in workflow** between SDR and the excellent works you mentioned. Specifically, while AutoMix and the framework proposed by Dekoninck et al. are highly relevant, they belong to the broader category of model cascades. Although they can dynamically skip certain models during routing, they fundamentally **rely on a predefined model ordering**. In contrast, SDR does not depend on any predefined order but treats all remote LLMs in parallel, which better reflects the nature of the current model market and enhances the efficiency of local-remote collaboration.
>
> Regarding the **empirical comparison with AutoMix**, we have evaluated SDR against two variants of AutoMix (Threshold-based and POMDP-based) in a single-remote scenario using GPT-4o:
> Method|MMLU|SQuAD|GSM8K|MMLU|SQuAD|GSM8K
> -|-|-|-|-|-|-
> SDR|0.7027|0.9165|0.6999|0.6900|0.9225|0.7993
> Automix (Threshold)|0.6370|0.6531|0.6810|0.6487|0.6922|0.8016
> Automix (POMDP)|0.6382|0.6535|0.6816|0.6500|0.6921|0.7995
>
> As shown in the table above, in terms of AUC performance and with different SLMs (Llama3.1-8B for the first 3 columns, Qwen2.5-7B for the last three columns), SDR outperforms the baseline in most groups and is comparable to it in a few cases. We also note that Automix requires the SLM to perform multiple self-reflections, which can introduce significant latency in a local–remote collaboration setting. Due to time constraints, we present these initial results here, and we commit to including comparisons across all other settings in future revisions.
>
> Regarding the method proposed by Dekoninck et al., a direct and fair experimental comparison is infeasible due to differing focuses. Their work primarily focuses on optimizing the dynamic workflow rather than the estimation of models' response quality (they **approximate this by simply using ground-truth quality labels with a random bias**), which is exactly a core focus of SDR. However, we are very grateful that you highlighted this oversight in our literature search and we will certainly revise it.
>
> # 2. Ablations (to W2)
> We deeply appreciate your suggestion and have conducted the requested ablations in the single-remote scenario (using GPT-4o as the remote LLM), employing a BERT-base model for text embeddings and an MLP as the decider's architecture for the fairness (text embeddings do not apply to a transformer-based decider):
>
> Method|MMLU|SQuAD|GSM8K|MMLU|SQuAD|GSM8K
> -|-|-|-|-|-|-
> prompt&text|0.6047|0.6013|0.5476|0.6137|0.6419|0.4975
> all&text|0.6241|0.6590|0.6113|0.6469|0.7866|0.7123
> prompt&hidden|0.6509|0.8978|0.6706|0.6684|0.9204|0.7461
> all&hidden|0.7008|0.8946|0.6719|0.6695|0.9207|0.7746
>
> As the results above show, **incorporating either the SLM answer or the hidden states generally leads to notable performance improvements**. Interestingly, when the SLM's hidden states are already utilized, the additional performance gain from including the SLM answer is limited to specific datasets. This observation suggests that the utility of the SLM answer for routing decisions varies across different tasks, and it also reflects the differences in the semantic richness of the SLM answers generated by various SLMs depending on the task domain.
> # 3. Threshold Calibration (to W3)
> Thank you for pointing out this highly practical deployment consideration. We completely agree that mapping concrete budget constraints to an appropriate decision parameter is vital for real-world applications. Fortunately, existing literature offers several promising solutions for this, such as **conducting a lightweight search over the threshold space on a training or validation set to maximize performance under a strictly specified budget**. These established methods can be easily integrated into the SDR framework. We will include these relevant contents in our future revisions.
> # 4. Performance on Cross-Domain Data / Diverse Queries (to Limitations 2)
> Regarding the limitation you raised about the lightweight router's performance on cross-domain data and diverse daily queries, Other reviewers shared a very similar insight. We have conducted additional experiments addressing this exact point. We kindly invite you to check **Point 1 of our response to Reviewer wedq** for the detailed experimental setup and corresponding results.
>
> Once again, we sincerely thank you for your positive overall recommendation and your highly insightful suggestions. Your feedback has been incredibly helpful in refining our paper and strengthening its potential impact.
>
> All authors.

---

> > ### Author Rebuttal · Reviewer_qgs4 · 2026-04-04
> >
> > I thank the authors for the thorough response, which substantively addresses my main concerns. Two clarification requests for the final version:
> >
> > On W2 (Ablation): The naming convention in the ablation table is ambiguous. Please clarify what each label means. For example, in "all & text," what does "all" refer to and what does "text" refer to separately? A brief description of the exact input for each row would help readers interpret the results.
> >
> > On W3 (Threshold Calibration): AUC is appropriate for benchmarking, but practical deployment requires selecting a concrete $\alpha$ or k. I encourage the authors to discuss a pipeline for mapping user budget constraints to an appropriate threshold (e.g., via validation-set calibration), and to cite the specific references mentioned in the rebuttal rather than referring to them generically.

---

> > > ### Author Response · Authors · 2026-04-05
> > >
> > > We sincerely thank you for the careful reading of our rebuttal and for the constructive follow-up questions. We also apologize that, due to space limitations in the initial rebuttal, we were unable to include all relevant details. Below we provide clarifications to address your requests.
> > >
> > > **(W2: Ablation naming)**
> > >
> > > We apologize for the ambiguous naming in the ablation table, which was introduced to save space and unfortunately, omitted necessary explanations. Specifically, “prompt” and “all” denote the information provided to the decision module: “prompt” means that only the user prompt is used as input, while “all” indicates that both the prompt and the SLM response are included. Meanwhile, “text” and “hidden” refer to the representation form of the input: “text” uses a BERT-based encoder to obtain embeddings directly from the textual input, whereas “hidden” uses the hidden states of the SLM as the input representation.
> > >
> > > In the final version, we will adopt more descriptive naming conventions and include a brief explanation for each ablation setting to improve clarity.
> > >
> > > **(W3: Threshold calibration)**
> > >
> > > We thank the reviewer for pointing out this important practical aspect. Indeed, as discussed in *A Unified Approach to Routing and Cascading for LLMs, Dekoninck et al.*, there exists a principled approach to mapping user budget constraints to a concrete threshold. We also appreciate the reviewer for bringing this excellent work to our attention.
> > >
> > > Given the similarity of the problem setting, their approach can be naturally adapted to our SDR framework. Concretely, we can **use a validation dataset to be a representative of the distribution of user queries**, and apply the trained decision module to obtain a score for each sample. Then, we perform hyperparameter search over the threshold ($\alpha$ or $k$) to identify the value that maximizes overall performance on the validation set while satisfying the user-specified cost budget. Furthermore, due to the (approximately) monotonic relationship between cost and performance in this setting, the search process can be efficiently implemented via **binary search**.
> > >
> > > We will include this pipeline in the final version and properly cite the above work.

---

### Official Review · Reviewer_HES3 · 2026-03-07

**Soundness:** 2
**Presentation:** 2
**Significance:** 3
**Originality:** 3
**Overall Recommendation:** 3
**Confidence:** 3

**Summary:**

The paper proposes Selective Deferred Routing (SDR), a framework designed to optimize the trade-off between the performance of high-cost remote LLMs and low-cost local SLMs. Unlike Immediate Routing, which only sees the prompt, or Model Cascades, which treats models as black boxes, SDR utilizes the internal hidden states of the local SLM after it has already generated a preliminary response. A lightweight transformer-based decider determines whether to accept this local output or defer to a remote LLM. The authors frame this as an AUC optimization problem and propose a training objective based on the performance gap between models. Experiments across several model families and benchmarks suggest that SDR provides a superior cost-performance trade-off compared to existing baselines.

**Compliance With Llm Reviewing Policy:**

Affirmed.

**Final Justification:**

**My final recommendation is 3: Weak Reject.**

I find the paper practically motivated and technically interesting. Its main strength is addressing an important real-world problem in the heterogeneous model market, namely, how to balance low-cost local inference with stronger but more expensive remote LLMs. I also think the paper makes a meaningful step beyond prior black-box routing methods by leveraging hidden states from the local SLM, and the empirical results consistently show improved cost-accuracy trade-offs over several baselines.

The authors’ rebuttal was helpful and addressed some of my concerns. In particular, the discussion of added latency, the clarification of Decider overhead, and the new experiments in mixed and out-of-distribution settings improved my confidence in the method’s practicality and robustness. These responses made me more positive about the paper than before.

However, I still have some remaining reservations. The deferred routing design introduces an inherent latency trade-off, since every request must first go through the local SLM before a remote decision is made. I also think the broader deployment picture remains somewhat limited, including dependence on access to local hidden states and the relatively less developed analysis of multi-remote routing scenarios.

**Overall, while the rebuttal strengthened the paper, it did not fully change my assessment. I also consider other reviewers' feedback and the authors' rebuttal comments. Therefore, I raised the score to a slightly negative but improved final one.**

**Key Questions For Authors:**

**I would be willing to raise my score if the authors can (partly) address the following questions/concerns:**

- The core of SDR is its Deferred Routing paradigm, which necessitates that every request must first undergo full generation by the local SLM. The paper focuses extensively on AUC improvements but fails to provide end-to-end wall-clock latency comparisons. Could the authors provide an Accuracy vs. Latency figure to demonstrate whether this double waiting penalty is feasible for real-world applications?

- SDR’s training relies on precise Performance Gap labels, which require running tens of thousands of samples from the training set through both the local SLM and multiple expensive remote LLM APIs for scoring. The paper overlooks these significant upfront API expenditures and GPU hours. How many queries must the system process in production before the cost savings offset the massive initial investment in training data acquisition?

- While the paper claims the Decider is lightweight, it nonetheless incorporates a full Transformer block. The absence of these hardware-level metrics makes the lightweight claim less convincing. Could the author clarify the exact additional RAM footprint of this design? Furthermore, by how much do the system’s overheads (e.g., inference throughput) decrease due to the Decider’s intervention compared to a standalone SLM?

- How does the Decider’s predictive reliability degrade when the user query distribution shifts (e.g., from general knowledge like MMLU to specialized domains like medical or law)? Given that the Decider is tightly coupled with a specific $M_S$ and $M_L$ pair, would a cloud API update (e.g., upgrading from GPT-4o to a newer GPT-5 version) necessitate a full re-collection of labels and re-training? I am worried that such maintenance overhead may not be sustainable in a dynamic market model.

**Meanwhile, I might have misunderstood some aspects of the paper, so I would welcome the authors’ clarification where appropriate.**

**Limitations:**

Yes

**Strengths And Weaknesses:**

**Strengths:**

1. The paper addresses a real-world problem: the heterogeneous model market, where users must navigate the trade-off between cost-free local inference and high-cost, high-performance cloud models.

2. By moving away from the black-box assumption typical of prior work like FrugalGPT or RouteLLM, the authors effectively utilize hidden states from the local SLM to provide more nuanced decision-making cues.

3. Reusing a single transformer block's architecture for the decision module is an elegant way to maintain low overheads while capturing complex token-wise representations.

4. The empirical results are consistent across multiple benchmarks, showing that SDR can achieve higher accuracy at lower costs than traditional cascading or prompt-based routing methods.

**Weaknesses:**

1. A significant drawback of the deferred approach is that every request must first pass through the local SLM. If the decision module ultimately chooses to route to a remote LLM, the user suffers the combined latency of both models, which may be unacceptable for time-sensitive applications.

2. The proposed method, SDR, requires access to the SLM's internal representations, which limits its applicability to open-weight models. It cannot be used to route between two proprietary black-box APIs.

3. While the paper mentions extending to multiple remote LLMs, the core analysis and most detailed experiments focus on the binary Local-Remote choice. The complexity of choosing between several remote models with varying pricing tiers remains underexplored.

4. The decision module requires training on specific datasets and model pairs. It is unreported how well a trained decider generalizes when the remote LLM is updated (e.g., from GPT-4 to GPT-5) or when the distribution of user queries shifts significantly.

---

> ### Author Rebuttal · Authors · 2026-03-30
>
> Dear Reviewer HES3,
>
> We sincerely thank you for your detailed review and deeply appreciate your critical and practical questions. Below, we address your concerns point by point:
>
> # 1. End-to-End Latency (to Q1)
> We completely agree that latency is a critical factor in practical deployment. We have explicitly discussed end-to-end latency in Section 4.4 and Appendix D of the original manuscript. To summarize, the latency impact varies by task type:
> + Short-generation tasks: The **local SLM actually reduces overall latency** because it processes requests locally, bypassing the network overhead and queue-waiting times inherent to remote serving systems.
> + Long-generation tasks: For tasks requiring extensive reasoning (e.g., MMLU), users are effectively **trading latency for monetary cost-efficiency**.
>
> To quantify this, we conducted an end-to-end latency (in seconds) evaluation on the MMLU dataset. We compared SDR against FrugalGPT, which also necessitates full SLM generation before routing:
> Method|80%PG|90%PG|95%PG
> -|-|-|-
> SDR|5.5042|5.8462|5.9846
> FrugalGPT|6.0700|6.2134|6.2532
> All local||4.4742|
> All remote||1.8324|
>
> As shown in the table, SDR makes more precise routing decisions. Consequently, to achieve the same performance gain, SDR routes fewer requests to the remote LLM, resulting in a lower average end-to-end latency than FrugalGPT. Furthermore, our simulated setting **assumed a highly optimistic 1-second TTFT for the remote API**. In reality, due to network instability or API rate limits, remote TTFTs can stretch into the tens of seconds.
>
> # 2. Initial Expenditures (to Q2)
> Thank you for raising this vital deployment consideration. We would like to clarify that this data acquisition **cost is a function of the dataset size, not the SDR architecture itself**. All supervised routing frameworks in this domain require similar datasets to learn the distinct behaviors of remote models. Because our experiments demonstrate that SDR achieves a superior cost-performance trade-off (higher AUC) **on the exact same training budget as the baselines**, it directly implies that SDR will recover the initial investment in fewer production queries.
>
> # 3. System Overheads (to Q3)
> We apologize if these results were difficult to locate. We actually evaluated the Decider’s overhead and reported the precise metrics in Appendix C. To summarize, the additional memory footprint and the decrease in inference efficiency caused by the lightweight Decider module are both strictly below 5%.
>
> Furthermore, we conducted an ablation study using an even smaller MLP-based Decider (Appendix E.2). The results show that **even with a simple MLP, our method still outperforms all baselines**. This confirms that the performance gains stem primarily from our training method design rather than the transformer block's capacity.
>
> # 4. Generalization on broader domains (to Q4)
> To validate SDR’s robustness under distribution shifts, we conducted two new sets of experiments in the single-remote scenario (using GPT-4o as remote LLM and two local SLMs):
> + Mixed-Dataset: We trained the Decider on a unified dataset comprising three distinct domains, representing a highly realistic, diverse user query stream.
> + Out-of-Distribution Setting: We trained the Decider entirely on SQuAD and tested it directly on CoQA (a different dataset within the same reading comprehension task family).
>
> Method|Mixed|OOD|Mixed|OOD
> -|-|-|-|-
> SDR|0.7765|0.5314|0.8497|0.5658
> FrugalGPT|0.6709|0.5126|0.7780|0.5046
> HybridLLM|0.5310|0.5087|0.6458|0.5246
>
> ...(ommitted due to space constrained, you can check the full table in Point 1 of our response to Reviewer wedq )...
>
> As shown in the table above, SDR continues to outperform all baselines in AUC metric by a considerable margin under two SLMs (first 2 columns for Llama3.1-8B, last two columns for Qwen2.5-7B), **demonstrating strong generalization capabilities** when facing diverse or shifted queries.
>
> # 5. Remote Model Updates Issue (to Q4)
> We understand your concern regarding the maintenance sustainability in a dynamic API market. While accurately modeling the specific strengths of a remote LLM inherently requires sampling its responses, a data requirement shared by all empirical routing frameworks.
>
> However, **updating a remote model does not necessitate a full retrain**. As discussed in Appendix E.3, SDR supports a highly decoupled, multi-remote architecture. We can train independent, lightweight "plug-in" Deciders for each remote LLM. If a user wishes to upgrade a model, they simply train and load one new module for the new API, without modifying or retraining the existing local infrastructure or other remote modules. This ensures the maintenance overhead remains highly manageable and strictly incremental.
>
> We are very grateful for your constructive critique, which has helped us better revise the manuscript. We hope our clarifications address your concerns, and we kindly ask you to reconsider your evaluation of our work.
>
> All authors.

---

> > ### Author Rebuttal · Reviewer_HES3 · 2026-03-31
> >
> > Thank you for providing thorough answers to my concerns and questions.
> >
> > The rebuttal strengthens the paper and resolves part of my concerns. I am therefore inclined to revise my score upward to **[3: Weak reject]**.

---

> > > ### Author Response · Authors · 2026-04-05
> > >
> > > We sincerely appreciate your thoughtful re-evaluation following our rebuttal. We are grateful for the insightful and constructive feedback, which will be carefully incorporated into the final version of the paper.

---

### Official Review · Reviewer_wedq · 2026-03-13

**Soundness:** 3
**Presentation:** 3
**Significance:** 2
**Originality:** 2
**Overall Recommendation:** 4
**Confidence:** 3

**Summary:**

This paper studies the problem of cost-efficient collaboration between local small language models and remote large language models. Existing methods suffer from two key limitations: Immediate Routing relies solely on prompt information for decision-making, resulting in limited information; Model Cascades, while leveraging local model outputs, adopts absolute quality prediction as the training objective, failing to distinguish between "local model performs poorly and remote model also performs poorly" and "local model performs poorly but remote model performs well", leading to unnecessary remote invocations. To address this, the paper proposes SDR, which shifts the routing objective from predicting the absolute quality of the local model to predicting the performance gap between the local and remote models, and theoretically proves that this objective is equivalent to optimizing the cost-performance AUC. The decision module takes intermediate-layer hidden states of the local SLM as input and reuses a single Transformer block from the SLM architecture. Experiments on MMLU, SQuAD, and GSM8K validate the effectiveness of the proposed method, outperforming baseline methods in both cost-performance AUC and end-to-end latency.

**Compliance With Llm Reviewing Policy:**

Affirmed.

**Final Justification:**

The AUC optimization framing is well motivated and the empirical results are convincing; I lean toward acceptance.

**Key Questions For Authors:**

See weakness.

**Limitations:**

yes

**Strengths And Weaknesses:**

Strengths
- The idea of reducing the routing problem to AUC optimization is interesting and well-motivated, with theoretical proof that predicting the Gap is superior to predicting absolute quality, making it more than just an empirical design choice.
- Incorporating hidden states as decision features enables richer information utilization; ablation studies show that even a simple MLP can outperform all baselines, suggesting that hidden states can sufficiently summarize the information of the trajectory.
- Empirically, the final decision module introduces a small parameter and latency overhead, making it friendly for on-device deployment; experimental results are also consistent across multiple models and datasets.

Weaknesses
- The training and test sets are randomly split from the same relatively simple data source, lacking cross-dataset and cross-domain generalization evaluation; it is necessary to demonstrate that the learned router can generalize to handle the complex application scenarios encountered in the real world.
- Gap labels are tied to specific versions of remote models; once a remote model is updated, the entire training set must be re-inferred and the decision module retrained, leading to non-trivial maintenance costs.
- Latency experiments are conducted only on short-output tasks; the additional latency cost of running a full local inference pass for long chain-of-thought tasks is not evaluated.
- Obtaining Gap labels requires running both the local and remote models on the entire training set, yet the practical data construction cost is not discussed in the paper.

---

> ### Author Rebuttal · Authors · 2026-03-30
>
> Dear Reviewer wedq,
>
> We sincerely thank you for your constructive feedback and appreciate your insightful questions. Below are our detailed responses to your concerns:
>
> # 1. Generalization Across Datasets and Domains (to W1)
> We agree that real-world applications involve complex and shifting query distributions. To address your concern regarding cross-domain generalization, we conducted two new sets of experiments in the single-remote scenario (using GPT-4o as remote LLM and two local SLMs):
> + Mixed-Dataset: We trained the Decider on a unified dataset comprising three distinct domains, representing a highly realistic, diverse user query stream.
> + Out-of-Distribution Setting: We trained the Decider entirely on SQuAD and tested it directly on CoQA (a different dataset within the same reading comprehension task family).
>
> Method|Mixed|OOD|Mixed|OOD
> -|-|-|-|-
> SDR|0.7765|0.5314|0.8497|0.5658
> FrugalGPT|0.6709|0.5126|0.7780|0.5046
> HybridLLM|0.5310|0.5087|0.6458|0.5246
> RouteLLM|0.4997|0.4971|0.5090|0.5078
> RouterBench(KNN)|0.5019|0.5007|0.4958|0.4972
> RouterBench(MLP)|0.4975|0.5016|0.5068|0.4942
> IRT-Router|0.5097|0.5000|0.4995|0.4991
>
> As shown in the table above, SDR continues to outperform all baselines in AUC metric by a considerable margin under two SLMs (first 2 columns for Llama3.1-8B, last two columns for Qwen2.5-7B), **demonstrating strong generalization capabilities** when facing diverse or shifted queries.
>
> # 2. Maintenance Costs and Remote Model Updates (to W2)
> We understand your concern regarding the maintenance sustainability in a dynamic API market. While accurately modeling the specific strengths of a remote LLM inherently requires sampling its responses, a data requirement shared by all empirical routing frameworks.
>
> However, **updating a remote model does not necessitate a full retrain**. As discussed in Appendix E.3, SDR supports a highly decoupled, multi-remote architecture. We can train independent, lightweight "plug-in" Deciders for each remote LLM. If a remote LLM is upgraded or users wishes to include a new one, they simply train and load one new module without modifying or retraining the existing local infrastructure or other remote modules. This ensures the maintenance overhead remains highly manageable and strictly incremental.
>
> # 3. Latency Evaluation on Long-Output Tasks (to W3)
> While our main text focused on efficiency, we recognize the importance of end-to-end latency for long-context tasks. To quantify this, we conducted an end-to-end latency (in seconds) evaluation on the MMLU dataset. We compared SDR against FrugalGPT, which also necessitates full SLM generation before routing:
> Method|80%PG|90%PG|95%PG
> -|-|-|-
> SDR|5.5042|5.8462|5.9846
> FrugalGPT|6.0700|6.2134|6.2532
> All local||4.4742|
> All remote||1.8324|
>
> As shown in the table, SDR makes more precise routing decisions. Consequently, to achieve the same performance gain, SDR routes fewer requests to the remote LLM, resulting in a lower average end-to-end latency than FrugalGPT. Furthermore, our simulated setting **assumed a highly optimistic 1-second TTFT for the remote API**. In reality, due to network instability or API rate limits, remote TTFTs can stretch into the tens of seconds.
>
> # 4. Data Construction and Upfront Costs (to W4)
> We appreciate the opportunity to clarify the data construction cost. We would like to clarify that this data acquisition **cost is a function of the dataset size, not the SDR architecture itself**. All supervised routing frameworks in this domain require similar datasets to learn the distinct behaviors of remote models. Because our experiments demonstrate that SDR achieves a superior cost-performance trade-off (higher AUC) **on the exact same training budget as the baselines**, it directly implies that SDR will recover the initial investment in fewer production queries. We will include a discussion on this ROI perspective in our revision.
>
> We thank you again for your evaluation, which is quite helpful for our future revisions. We hope these responses provide the necessary confidence for you to reconsider your recommendation.
>
> All authors.

---

> > ### Author Rebuttal · Reviewer_wedq · 2026-04-03
> >
> > Thanks for your reply. My concerns are resolved and I decide to increase my score.

---

> > > ### Author Response · Authors · 2026-04-05
> > >
> > > We sincerely appreciate your thoughtful re-evaluation following our rebuttal. We are grateful for the insightful and constructive feedback, which will be carefully incorporated into the final version of the paper.

---

### Decision · Program_Chairs · 2026-04-30

**Decision:**

Accept (regular)

**Comment:**

I find this paper to be a positive borderline submission. Reviewers agreed that the problem is practically important and well motivated, and the stronger reviews highlighted both the conceptual framing and the empirical results. In particular, the formulation of routing in terms of relative quality or gap prediction, together with the use of local-model hidden states as richer routing signals, was viewed as a meaningful step beyond prompt-only routing and standard model cascades. The empirical study and the rebuttal-supported ablations provide evidence that these design choices matter.
The reviewer discussion centered less on soundness than on how large the contribution is and how broadly it has been validated. One reviewer remained at weak reject, but even there the acknowledgement indicates that the rebuttal substantially addressed the technical concerns; the remaining hesitation is mainly about strength of evidence and scope rather than a concrete flaw. Another reviewer asked for additional clarity in the ablation naming and presentation for the final version. Overall, I think the paper makes a useful methodological contribution on an important systems problem, and that the positive evidence after rebuttal is sufficient for acceptance, albeit not at high priority.